# Margin-Based Generalization Lower Bounds for Boosted Classifiers

Allan Grønlund [‡§]     Lior Kamma [§]     Kasper Green Larsen [§]     Alexander Mathiasen [§]

Jelani Nelson [*]

## Abstract

Boosting is one of the most successful ideas in machine learning. The most well-accepted explanations for the low generalization error of boosting algorithms such as AdaBoost stem from margin theory. The study of margins in the context of boosting algorithms was initiated by Schapire, Freund, Bartlett and Lee (1998) and has inspired numerous boosting algorithms and generalization bounds. To date, the strongest known generalization (upper bound) is the $k$th margin bound of Gao and Zhou (2013). Despite the numerous generalization upper bounds that have been proved over the last two decades, nothing is known about the tightness of these bounds. In this paper, we give the first margin-based lower bounds on the generalization error of boosted classifiers. Our lower bounds nearly match the $k$th margin bound and thus almost settle the generalization performance of boosted classifiers in terms of margins.

## 1 Introduction

*Boosting algorithms* produce highly accurate classifiers by combining several less accurate classifiers and are amongst the most popular learning algorithms, obtaining state-of-the-art performance on several benchmark machine learning tasks [KMF+17, CG16]. The most famous of these boosting algorithm is arguably AdaBoost [FS97]. For binary classification, AdaBoost takes a training set $S = \langle (x_1, y_1), \ldots, (x_m, y_m) \rangle$ of $m$ labeled samples as input, with $x_i \in \mathcal{X}$ and labels $y_i \in \{-1, 1\}$. It then produces a classifier $f$ in iterations: in the $j$th iteration, a base classifier $h_j : \mathcal{X} \to \{-1, 1\}$ is trained on a reweighed version of $S$ that emphasizes data points that $f$ struggles with and this classifier is then added to $f$. The final classifier is obtained by taking the sign of $f(x) = \sum_j \alpha_j h_j(x)$, where the $\alpha_j$'s are non-negative coefficients carefully chosen by AdaBoost. The base classifiers $h_j$ all come from a *hypothesis set* $\mathcal{H}$, e.g. $\mathcal{H}$ could be a set of small decision trees or similar. As AdaBoost's training progresses, more and more base classifiers are added to $f$, which in turn causes the training error of $f$ to decrease. If $\mathcal{H}$ is rich enough, AdaBoost will eventually classify all the data points in the training set correctly [FS97].

Early experiments with AdaBoost report a surprising generalization phenomenon [SFBL98]. Even after perfectly classifying the entire training set, further iterations keeps improving the test accuracy. This is contrary to what one would expect, as $f$ gets more complicated with more iterations, and thus prone to overfitting. The most prominent explanation for this phenomena is margin theory, introduced by Schapire *et al.* [SFBL98]. The margin of a training point $(x_i, y_i)$ is a number in $[-1, 1]$, which can be interpreted, loosely speaking, as the classifier's confidence on that point. Formally, we say that $f(x) = \sum_j \alpha_j h_j(x)$ is a *voting classifier* if $\alpha_j \geq 0$ for all $j$. Note that one can additionally assume

---

[‡] All authors contributed equally, and are presented in alphabetical order.

[§] Department of Computer Science, Aarhus University, {`jallan,lior.kamma,larsen,alexmath`}`@cs.au.dk`

[*] Department of EECS, UC Berkeley, `minilek@berkeley.edu`

without loss of generality that $\sum_j \alpha_j = 1$ since normalizing each $\alpha_i$ by $\sum_j \alpha_j$ leaves the sign of $f(x_i)$ unchanged. The margin of a point $(x_i, y_i)$ with respect to a voting classifier $f$ is then defined as

$$\text{margin}(x_i) := y_i f(x_i) \quad = \quad y_i \sum_j \alpha_j h_j(x_i) .$$

Thus $\text{margin}(x_i) \in [-1, 1]$, and if $\text{margin}(x_i) > 0$, then taking the sign of $f(x_i)$ correctly classifies $(x_i, y_i)$. Informally speaking, margin theory guarantees that voting classifiers with large (positive) margins have a smaller generalization error. Experimentally AdaBoost has been found to continue to improve the margins even when training past the point of perfectly classifying the training set. Margin theory may therefore explain the surprising generalization phenomena of AdaBoost. Indeed, the original paper by Schapire *et al.* [SFBL98] that introduced margin theory, proved the following margin-based generalization bound. Let $\mathcal{D}$ be an unknown distribution over $\mathcal{X} \times \{-1, 1\}$ and assume that the training data $S$ is obtained by drawing $m$ i.i.d. samples from $\mathcal{D}$. Then with high probability over $S$ it holds that for every margin $\theta \in (0, 1]$, *every* voting classifier $f$ satisfies

$$\Pr_{(x,y) \sim \mathcal{D}}[yf(x) \leq 0] \leq \Pr_{(x,y) \sim S}[yf(x) < \theta] + O\left(\sqrt{\frac{\ln|\mathcal{H}|\ln m}{\theta^2 m}}\right). \tag{1}$$

The left-hand side of the equation is the out-of-sample error of $f$ (since $\text{sign}(f(x)) \neq y$ precisely when $yf(x) < 0$). On the right-hand side, we use $(x, y) \sim S$ to denote a uniform random point from $S$. Hence $\Pr_{(x,y) \sim S}[yf(x) < \theta]$ is the fraction of training points with margin less than $\theta$. The last term is increasing in $|\mathcal{H}|$ and decreasing in $\theta$ and $m$. Here it is assumed $\mathcal{H}$ is finite. A similar bound can be proved for infinite $\mathcal{H}$ by replacing $|\mathcal{H}|$ by $d \lg m$, where $d$ is the VC-dimension of $\mathcal{H}$. This holds for all the generalization bounds below as well. The generalization bound thus shows that $f$ has low out-of-sample error if it attains large margins on most training points. This fits well with the observed behaviour of AdaBoost in practice.

The generalization bound above holds for every voting classifier $f$, i.e. regardless of how $f$ was obtained. Hence a natural goal is to design boosting algorithms that produce voting classifiers with large margins on many points. This has been the focus of a long line of research and has resulted in numerous algorithms with various margin guarantees, see e.g. [GS98, Bre99, BDST00, RW02, RW05, GLM19]. One of the most well-known of these is Breimann's ArcGV [Bre99]. ArcGV produces a voting classifier maximizing the *minimal* margin, i.e. it produces a classifier $f$ for which $\min_{(x,y) \in S} yf(x)$ is as large as possible. Breimann complemented the algorithm with a generalization bound stating that with high probability over the sample $S$, it holds that every voting classifier $f$ satisfies:

$$\Pr_{(x,y) \sim \mathcal{D}}[yf(x) \leq 0] \leq O\left(\frac{\ln|\mathcal{H}|\ln m}{\hat{\theta}^2 m}\right), \tag{2}$$

where $\hat{\theta} = \min_{(x,y) \in S} yf(x)$ is the minimal margin over all training examples. Notice that if one chooses $\theta$ as the minimal margin in the generalization bound (1) of Schapire *et al.* [SFBL98], then the term $\Pr_{(x,y) \sim S}[yf(x) < \theta]$ becomes 0 and one obtains the bound

$$\Pr_{(x,y) \sim \mathcal{D}}[yf(x) \leq 0] \leq O\left(\sqrt{\frac{\ln|\mathcal{H}|\ln m}{\hat{\theta}^2 m}}\right),$$

which is weaker than Breimann's bound and motivated his focus on maximizing the minimal margin. Minimal margin is however quite sensitive to outliers and work by Gao and Zhou [GZ13] proved a generalization bound which provides an interpolation between (1) and (2). Their bound is known as the $k$th margin bound, and states that with high probability over the sample $S$, it holds for every margin $\theta \in (0, 1]$ and every voting classifier $f$ that:

$$\Pr_{(x,y) \sim \mathcal{D}}[yf(x) < 0] \leq \Pr_{(x,y) \sim S}[yf(x) < \theta] + O\left(\frac{\ln|\mathcal{H}|\ln m}{\theta^2 m} + \sqrt{\Pr_{(x,y) \sim S}[yf(x) < \theta]\frac{\ln|\mathcal{H}|\ln m}{\theta^2 m}}\right).$$

The $k$th margin bound remains the strongest margin-based generalization bound to date (see Section 1.2 for further details). The $k$th margin bound recovers Breimann's minimal margin bound by choosing $\theta$ as the minimal margin (making $\Pr_{(x,y) \sim S}[yf(x) < \theta] = 0$), and it is always at most the

same as the bound (1) by Schapire *et al.* As with previous generalization bounds, it suggests that boosting algorithms should focus on obtaining a large margin on as large a fraction of training points as possible.

Despite the decades of progress on generalization *upper* bounds, we still do not know how tight these bounds are. That is, we do not have any margin-based generalization *lower* bounds. Generalization lower bounds are not only interesting from a theoretical point of view, but also from an algorithmic point of view: If one has a provably tight generalization bound, then a natural goal is to design a boosting algorithm minimizing a loss function that is equal to this generalization bound. This approach makes most sense with a matching lower bound as the algorithm might otherwise minimize a sub-optimal loss function. Furthermore, a lower bound may also inspire researchers to look for other parameters than margins when explaining the generalization performance of voting classifiers. Such new parameters may even prove useful in designing new algorithms, with even better generalization performance in practice.

## 1.1 Our Results

In this paper we prove the first margin-based generalization lower bounds for voting classifiers. Our lower bounds almost match the $k$th margin bound and thus essentially settles the generalization performance of voting classifiers in terms of margins.

To present our main theorems, we first introduce some notation. For a ground set $\mathcal{X}$ and hypothesis set $\mathcal{H}$, let $C(\mathcal{H})$ denote the family of all voting classifiers over $\mathcal{H}$, i.e. $C(\mathcal{H})$ contains all functions $f : \mathcal{X} \to [-1, 1]$ that can be written as $f(x) = \sum_{h \in \mathcal{H}} \alpha_h h(x)$ such that $\alpha_h \geq 0$ for all $h$ and $\sum_h \alpha_h = 1$. For a (randomized) learning algorithm $\mathcal{A}$ and a sample $S$ of $m$ points, let $f_{\mathcal{A},S}$ denote the (possibly random) voting classifier produced by $\mathcal{A}$ when given the sample $S$ as input. With this notation, our first main theorem is the following:

**Theorem 1.** *For every large enough integer $N$, every $\theta \in (1/N, 1/40)$ and every $\tau \in [0, 49/100]$ there exist a set $\mathcal{X}$ and a hypothesis set $\mathcal{H}$ over $\mathcal{X}$, such that $\ln |\mathcal{H}| = \Theta(\ln N)$ and for every $m = \Omega\left(\theta^{-2} \ln |\mathcal{H}|\right)$ and for every (randomized) learning algorithm $\mathcal{A}$, there exist a distribution $\mathcal{D}$ over $\mathcal{X} \times \{-1, 1\}$ and a voting classifier $f \in C(\mathcal{H})$ such that with probability at least $1/100$ over the choice of samples $S \sim \mathcal{D}^m$ and the random choices of $\mathcal{A}$*

1. $\displaystyle \Pr_{(x,y) \sim S}[yf(x) < \theta] \leq \tau$*; and*

2. $\displaystyle \Pr_{(x,y) \sim \mathcal{D}}[yf_{\mathcal{A},S}(x) < 0] \geq \tau + \Omega\left( \frac{\ln |\mathcal{H}|}{m\theta^2} + \sqrt{\tau \cdot \frac{\ln |\mathcal{H}|}{m\theta^2}} \right).$

Theorem 1 states that for any algorithm $\mathcal{A}$, there is a distribution $\mathcal{D}$ for which the out-of-sample error of the voting classifier produced by $\mathcal{A}$ is at least that in the second point of the theorem. At the same time, one can find a voting classifier $f$ obtaining a margin of at least $\theta$ on at least a $1 - \tau$ fraction of the sample points. Our proof of Theorem 1 not only shows that such a classifier exists, but also provides an algorithm that constructs such a classifier. Loosely speaking, the first part of the theorem reflects on the nature of the distribution $\mathcal{D}$ and the hypothesis set $\mathcal{H}$. Intuitively it means that the distribution is not too hard and the hypothesis set is rich enough, so that it is possible to construct a voting classifier with good empirical margins. Clearly, we cannot hope to prove that the algorithm $\mathcal{A}$ constructs a voting classifier that has a margin of at least $\theta$ on a $1 - \tau$ fraction of the sample set, since we make no assumptions on the algorithm. For example, if the constant hypothesis $h_1$ that always outputs 1 is in $\mathcal{H}$, then $\mathcal{A}$ could be the algorithm that simply outputs $h_1$. The interpretation is thus: $\mathcal{D}$ and $\mathcal{H}$ allow for an algorithm $\mathcal{A}$ to produce a voting classifier $f$ with margin at least $\theta$ on a $1 - \tau$ fraction of samples. The second part of the theorem thus guarantees that regardless of which voting classifier $\mathcal{A}$ produces, it still has large out-of-sample error. This implies that every algorithm that constructs a voting classifier by minimizing the empirical risk, must have a large error. Formally, Theorem 1 implies that if $\Pr_{(x,y) \sim S}[yf_{\mathcal{A},S}(x) > \theta] \leq \tau$ then

$$\Pr_{(x,y) \sim \mathcal{D}}[yf_{\mathcal{A},S}(x) < 0] \geq \Pr_{(x,y) \sim S}[yf_{\mathcal{A},S}(x) > \theta] + \Omega\left( \frac{\ln |\mathcal{H}|}{m\theta^2} + \sqrt{\tau \cdot \frac{\ln |\mathcal{H}|}{m\theta^2}} \right) .$$

The first part of the theorem ensures that the condition is not void. That is, there exists an algorithm $\mathcal{A}$ for which $\Pr_{(x,y) \sim S}[yf_{\mathcal{A},S}(x) < \theta] \leq \tau$. Comparing Theorem 1 to the $k$th margin bound, we

see that the parameter $\tau$ corresponds to $\Pr_{(x,y)\sim S}[yf(x) < \theta]$. The magnitude of the out-of-sample error in the second point in the theorem thus matches that of the $k$th margin bound, except for a factor $\ln m$ in the first term inside the $\Omega(\cdot)$ and a $\sqrt{\ln m}$ factor in the second term. If we consider the range of parameters $\theta, \tau, \ln |\mathcal{H}|$ and $m$ for which the lower bound applies, then these ranges are almost as tight as possible. For $\tau$, note that the theorem cannot generally be true for $\tau > 1/2$, as the algorithm $\mathcal{A}$ that outputs a uniform random choice of hypothesis among $h_1$ and $h_{-1}$ (the constant hypothesis outputting $-1$), gives a (random) voting classifier $f_{\mathcal{A},S}$ with an expected out-of-sample error of $1/2$. This is less than the second point of the theorem would state if it was true for $\tau > 1/2$. For $\ln |\mathcal{H}|$, observe that our theorem holds for arbitrarily large values of $|\mathcal{H}|$. That is, the integer $N$ can be as large as desired, making $\ln |\mathcal{H}| = \Theta(\ln N)$ as large as desired. Finally, for the constraint on $m$, notice again that the theorem simply cannot be true for smaller values of $m$ as then the term $\ln |\mathcal{H}|/(m\theta^2)$ exceeds 1.

Our second main result gets even closer to the $k$th margin bound:

**Theorem 2.** *For every large enough integer $N$, every $\theta \in (1/N, 1/40)$, $\tau \in [0, 49/100]$ and every $m = \left(\theta^{-2} \ln N\right)^{1+\Omega(1)}$, there exist a set $\mathcal{X}$, a hypothesis set $\mathcal{H}$ over $\mathcal{X}$ and a distribution $\mathcal{D}$ over $\mathcal{X} \times \{-1, 1\}$ such that $\ln |\mathcal{H}| = \Theta(\ln N)$ and with probability at least $1/100$ over the choice of samples $S \sim \mathcal{D}^m$ there exists a voting classifier $f_S \in C(\mathcal{H})$ such that*

1. $\displaystyle \Pr_{(x,y)\sim S}[yf_S(x) < \theta] \leq \tau$*; and*

2. $\displaystyle \Pr_{(x,y)\sim \mathcal{D}}[yf_S(x) < 0] \geq \tau + \Omega\left(\frac{\ln |\mathcal{H}| \ln m}{m\theta^2} + \sqrt{\tau \cdot \frac{\ln |\mathcal{H}|}{m\theta^2}}\right)$.

Observe that the second point of Theorem 2 has an additional $\ln m$ factor on the first term in $\Omega(\cdot)$ compared to Theorem 1. It is thus only off from the $k$th margin bound by a $\sqrt{\ln m}$ factor in the second term and hence completely matches the $k$th margin bound for small values of $\tau$. To obtain this strengthening, we replaced the guarantee in Theorem 1 saying that *all* algorithms $\mathcal{A}$ have such a large out-of-sample error. Instead, Theorem 2 demonstrates only the existence of a voting classifier $f_S$ (that is chosen as a function of the sample $S$) that simultaneously achieves a margin of at least $\theta$ on a $1 - \tau$ fraction of the sample points, and yet has out-of-sample error at least that in point 2. Since the $k$th margin bound holds with high probability *for all* voting classifiers, Theorem 2 rules out any strengthening of the $k$th margin bound, except for possibly a $\sqrt{\ln m}$ factor on the second additive term. Again, our lower bound holds for almost the full range of parameters of interest. As for the bound on $m$, our proof assumes $m = \left(\theta^{-2} \ln N\right)^{1+1/8}$, however the theorem holds for any constant greater than 1 in the exponent.

Finally, we mention that both our lower bounds are proved for a finite hypothesis set $\mathcal{H}$. This only makes the lower bounds stronger than if we proved it for an infinite $\mathcal{H}$ with bounded VC-dimension, since the VC-dimension of a finite $\mathcal{H}$, is no more than $\lg |\mathcal{H}|$.

## 1.2 Related Work

We mentioned above that the $k$th margin bound is the strongest margin-based generalization bound to date. Technically speaking, it is incomparable to the so-called *emargin* bound by Wang *et al.* [WSJ+11]. The $k$th margin bound by Gao and Zhou [GZ13], the minimum margin bound by Breimann [Bre99] and the bound by Schapire *et al.* [SFBL98] all have the form $\Pr_{(x,y)\sim \mathcal{D}}[yf(x) < 0] \leq \Pr_{(x,y)\sim S}[yf(x) < \theta] + \Gamma(\theta, m, |\mathcal{H}|, \Pr_{(x,y)\sim S}[yf(x) < \theta])$ for some function $\Gamma$. The emargin bound has a different (and quite involved) form, making it harder to interpret and compute. We will not discuss it in further detail here and just remark that our results show that for generalization bounds of the form studied in most previous work [SFBL98, Bre99, GZ13], one cannot hope for much stronger upper bounds than the $k$th margin bound.

## 2 Proof Overview

The main argument that lies in the heart of both proofs is a probabilistic method argument. With every labeling $\ell \in \{-1, 1\}^u$ we associate a distribution $\mathcal{D}_\ell$ over $\mathcal{X} \times \{-1, 1\}$. We then show that with some positive probability if we sample $\ell \in \{-1, 1\}^u$, $\mathcal{D}_\ell$ satisfies the requirements of Theorem 1

(respectively Theorem 2). We thus conclude the existence of a suitable distribution. We next give a more detailed high-level description of the proof for Theorem 1. The proof of Theorem 2 follows similar lines.

**Constructing a Family of Distributions.**  We start by first describing the construction of $\mathcal{D}_\ell$ for $\ell \in \{-1, 1\}^u$. Our construction combines previously studied distribution patterns in a subtle manner.

Ehrenfeucht *et al.* [EHKV89] observed that if a distribution $\mathcal{D}$ assigns each point in $\mathcal{X}$ a fixed (yet unknown) label, then, loosely speaking, every classifier $f$, that is constructed using only information supplied by a sample $S$, cannot do better than random guessing the labels for the points in $\mathcal{X} \setminus S$. Intuitively, consider a uniform distribution $\mathcal{D}_\ell$ over $\mathcal{X}$. If we assume, for example, that $|\mathcal{X}| \geq 10m$, then with very high probability over a sample $S$ of $m$ points, many elements of $\mathcal{X}$ are not in $S$. Moreover, assume that $\mathcal{D}_\ell$ associates every $x \in \mathcal{X}$ with a unique "correct" label $\ell(x)$. Consider some (perhaps random) learning algorithm $\mathcal{A}$, and let $f_{\mathcal{A},S}$ be the classifier it produces given a sample $S$ as input. If $\ell$ is chosen randomly, then, loosely speaking, for every point $x$ not in the sample, $f_{\mathcal{A},S}(x)$ and $\ell(x)$ are independent, and thus $\mathcal{A}$ returns the wrong label with probability $1/2$. In turn, this implies that there exists a labeling $\ell$ such that $\mathcal{A}$ is wrong on a constant fraction of $\mathcal{X}$ when receiving a sample $S \sim \mathcal{D}_\ell^m$. While the argument above can in fact be used to prove an arbitrarily large generalization error, it requires $|\mathcal{X}|$ to be large, and specifically to increase with $m$. This conflicts with the first point in Theorem 1, that is, we have to argue that a voting classifier $f$ with good margins exist for the sample $S$. If $S$ consists of $m$ distinct points, and each point in $\mathcal{X}$ can have an arbitrary label, then intuitively $\mathcal{H}$ needs to be very large to ensure the existence of $f$. In order to overcome this difficulty, we set $\mathcal{D}_\ell$ to assign very high probability to one designated point in $\mathcal{X}$, and the rest of the probability mass is then equally distributed between all other points. The argument above still applies for the subset of small-probability points. More precisely, if $\mathcal{D}_\ell$ assigns all but one point in $\mathcal{X}$ probability $\frac{1}{10m}$, then the expected generalization error (over the choice of $\ell$) is still $\Omega\left(\frac{1}{10m}|\mathcal{X}|\right)$. It remains to determine how large can we set $|\mathcal{X}|$. In the notations of the theorem, in order for a hypothesis set $\mathcal{H}$ to satisfy $\ln|\mathcal{H}| = \Theta(\ln N)$, and at the same time, have an $f \in C(\mathcal{H})$ obtaining margins of $\theta$ on most points in a sample, our proof (and specifically Lemma 3, described hereafter) requires $\mathcal{X}$ to be not significantly larger than $\frac{\ln N}{\theta^2}$, and therefore the generalization error we get is $\Omega\left(\frac{\ln|\mathcal{H}|}{\theta^2 m}\right)$. This accounts for the first term inside the $\Omega$-notation in the second point of Theorem 1.

Anthony and Bartlett [AB09, Chapter 5] additionally observed that for a distribution $\mathcal{D}$ that assigns each point in $\mathcal{X}$ a random label, if $S$ does not sample a point $x$ enough times, any classifier $f$, that is constructed using only information supplied by $S$, cannot determine with good probability the Bayes label of $x$, that is, the label of $x$ that minimizes the error probability. Intuitively, consider once more a distribution $\mathcal{D}_\ell$ that is uniform over $\mathcal{X}$. However, instead of associating every point $x \in \mathcal{X}$ with one correct label $\ell(x)$, $\mathcal{D}_\ell$ is now only slightly biased towards $\ell$. That is, given that $x$ is sampled, the label in the sample point is $\ell(x)$ with probability that is a little larger than $1/2$, say $(1 + \alpha)/2$ for some small $\alpha \in (0, 1)$. Note that every classifier $f$ has an error probability of at least $(1 - \alpha)/2$ on every given point in $\mathcal{X}$. Consider once again a learning algorithm $\mathcal{A}$ and the voting classifier $f_{\mathcal{A},S}$ it constructs. Loosely speaking, if $S$ does not sample a point $x$ enough times, then with good probability $f_{\mathcal{A},S}(x) \neq \ell(x)$. More formally, in order to correctly assign the Bayes label of $x$, an algorithm must see $\Omega(\alpha^{-2})$ samples of $x$. Therefore if we set the bias $\alpha$ to be $\sqrt{|\mathcal{X}|/(10m)}$, then with high probability the algorithm does not see a constant fraction of $\mathcal{X}$ enough times to correctly assign their label. In turn, this implies an expected generalization error of $(1 - \alpha)/2 + \Omega(\sqrt{|\mathcal{X}|/m})$, where the expectation is over the choice of $\ell$. By once again letting $|\mathcal{X}| = \frac{\ln N}{\theta^2}$ we conclude that there exists a labeling $\ell$ such that for $S \sim \mathcal{D}_\ell^m$, the expected generalization error of $f_{\mathcal{A},S}$ is $\frac{1-\alpha}{2} + \Omega\left(\sqrt{\frac{\ln|\mathcal{H}|}{\theta^2 m}}\right)$.

This expression is almost the second term inside the $\Omega$-notation in the theorem statement, though slightly larger. We note, however, for large values of $m$, the in-sample error is arbitrarily close to $1/2$. One challenge is therefore to reduce the in-sample-error, and moreover guarantee that we can find a voting classifier $f$ where the $(m\tau)$'th smallest margin for $f$ is at least $\theta$, where $\tau, \theta$ are the parameters provided by the theorem statement.

To this end, our proof subtly weaves the two ideas described above and constructs a family of distributions $\{\mathcal{D}_\ell\}_{\ell \in \{-1,1\}^u}$. Informally, we partition $\mathcal{X}$ into two disjoint sets, and conditioned on the sample point $x \in \mathcal{X}$ belonging to each of the subsets, $\mathcal{D}_\ell$ is defined similarly to be one of the two distribution patterns defined above. The main difficulty lies in delicately balancing all ingredients and

ensuring that we can find an $f$ with margins of at least $\theta$ on all but $\tau m$ of the sample points, while still enforcing a large generalization error. Our proof refines the proof given by Ehrenfeucht *et al.* and Anthony and Bartlett and shows that not only does there exists a labeling $\ell$ such that $f_{\mathcal{A},S}$ has large generalization error with respect to $\mathcal{D}_\ell$ (with probability at least $1/100$ over the randomness of $\mathcal{A}, S$), but rather that a large (constant) fraction of labelings $\ell$ share this property. This distinction becomes crucial in the proof.

**Small yet Rich Hypothesis Sets.** The technical crux in our proofs is the construction of an appropriate hypothesis set. Loosely speaking, the size of $\mathcal{H}$ has to be small, and most importantly, independent of the size $m$ of the sample set. On the other hand, the set of voting classifiers $C(\mathcal{H})$ is required to be rich enough to, intuitively, contain a classifier that with good probability has good in-sample margins for a sample $S \sim \mathcal{D}_\ell^m$ with a large fraction of labelings $\ell \in \{-1, 1\}^u$. Our main technical lemma presents a distribution $\mu$ over small hypothesis sets $\mathcal{H} \subset \mathcal{X} \to \{-1, 1\}$ such that for every *sparse* $\ell \in \{-1, 1\}^u$, that is $\ell_i = -1$ for a small number of entries $i \in [u]$, with high probability over $\mathcal{H} \sim \mu$, there exists some voting classifier $f \in C(\mathcal{H})$ that has minimum margin $\theta$ with $\ell$ over the entire set $\mathcal{X}$. In fact, the size of the hypothesis set does not depend on the size of $\mathcal{X}$, but only on the sparsity parameter $d$. More formally, we show the following.

**Lemma 3.** *For every $\theta \in (0, 1/40)$, $\delta \in (0, 1)$ and integers $d \leq u$, there exists a distribution $\mu = \mu(u, d, \theta, \delta)$ over hypothesis sets $\mathcal{H} \subset \mathcal{X} \to \{-1, 1\}$, where $\mathcal{X}$ is a set of size $u$, such that the following holds for $N = \Theta\left( \theta^{-2} \ln d \ln(\theta^{-2} d \delta^{-1}) e^{\Theta(\theta^2 d)} \right)$.*

1. *For all $\mathcal{H} \in \mathrm{supp}(\mu)$, we have $|\mathcal{H}| = N$; and*

2. *For every labeling $\ell \in \{-1, +1\}^u$, if no more than $d$ points $x \in \mathcal{X}$ satisfy $\ell(x) = -1$, then*

$$\Pr_{\mathcal{H} \sim \mu}\left[ \exists f \in \mathcal{C}(\mathcal{H}) : \forall x \in \mathcal{X}.\ \ell(x) f(x) \geq \theta \right] \geq 1 - \delta \ ,$$

In fact, we prove that if $\mathcal{H}$ is a random hypothesis set that also contains the hypothesis mapping all points to $1$, then with good probability $\mathcal{H}$ satisfies the second requirement in the theorem.

To show the existence of a good voting classifier in $C(\mathcal{H})$ our proof actually employs a slight variant of the celebrated AdaBoost algorithm, and shows that with high probability (over the choice of the random hypothesis set $\mathcal{H}$), the voting classifier constructed by this algorithm attains minimum margin at least $\theta$ over the entire set $\mathcal{X}$.

**Existential Lower Bound.** The difference between the generalization lower bound (second point) in Theorem 1 and 2 is a $\ln m$ factor in the first term inside the $\Omega(\cdot)$ notation. This term originated from having $\ln |\mathcal{H}|/\theta^2$ points with a probability mass of $1/10m$ in $\mathcal{D}_\ell$ and one point having the remaining probability mass. In the proof of Theorem 2, we first exploit that we are proving an existential lower bound by assigning all points the same label $1$. Since we are not proving a lower bound for every algorithm, this will not cause problems. We then change $|\mathcal{X}|$ to about $m/\ln m$ and assign each point the same probability mass $\ln m/m$ in the distribution $\mathcal{D}$. The key observation is that on a random sample $S$ of $m$ points, by a coupon-collector argument, there will still be $m^{\Omega(1)}$ points from $\mathcal{X}$ that were not sampled. From Lemma 3, we can now find a voting classifier $f$, such that $\mathrm{sign}(f(x))$ is $1$ on all points in $x \in S$, and $-1$ on a set of $d = \ln |\mathcal{H}|/\theta^2$ points in $\mathcal{X} \setminus S$. This means that $f$ has out-of-sample error $\Omega(d \ln m/m) = \Omega(\frac{\ln |\mathcal{H}| \ln m}{\theta^2 m})$ under distribution $\mathcal{D}$ and obtains a margin of $\theta$ on all points in the sample $S$.

# 3  Proof of Algorithmic Lower Bound

In this section we prove Theorem 1 assuming Lemma 3. The proof of Lemma 3, as well as the proof of Theorem 2 are deferred to the full version of the paper [GKL+19]. To prove Theorem 1, fix some integer $N$, and fix $\theta \in (1/N, 1/40)$. In order to ensure that the hypothesis set constructed using Lemma 3 is small enough, and specifically has size $N^{O(1)}$, we need the sparsity parameter to be not much larger than $\frac{\ln N}{\theta^2}$. As described in Section 2, the family of distributions we present will be defined separately over two subsets of $\mathcal{X}$. To this end, and for ease of notations, we let the size $u$ of $\mathcal{X}$ be $\frac{2 \ln N}{\theta^2}$, and let $d = \frac{\ln N}{\theta^2} = \frac{u}{2}$ be the size of each half. Finally, denote $\mathcal{X} = \{\xi_1, \dots, \xi_u\}$.

We start by constructing the family $\{\mathcal{D}_\ell\}_{\ell \in \{-1,1\}^u}$ of distributions over $\mathcal{X} \times \{-1, 1\}$. Fixing a labeling $\ell \in \{-1, 1\}^u$, we define $\mathcal{D}_\ell$ separately for the first $u/2$ points and the last $u/2$ points of $\mathcal{X}$. Intuitively, every point in $\{\xi_i\}_{i \in [u/2]}$ has a fixed label determined by $\ell$, however all points but one have a very small probability of being sampled according to $\mathcal{D}_\ell$. Every point in $\{\xi_i\}_{i \in [u/2+1,u]}$, on the other hand, has an equal probability of being sampled, however its label is not fixed by $\ell$, but instead slightly biased towards $\ell$. Formally, let $\alpha, \beta, \varepsilon \in [0, 1]$ be constants to be fixed later. We construct $\mathcal{D}_\ell$ using the ideas described earlier in Section 2, by sewing them together over two parts of the set $\mathcal{X}$. We assign probability $1 - \beta$ to $\{\xi_i\}_{i \in [u/2]}$ and $\beta$ to $\{\xi_i\}_{i \in [u/2+1,u]}$. That is, for $(x, y) \sim \mathcal{D}_\ell$, the probability that $x \in \{\xi_i\}_{i \in [u/2]}$ is $1 - \beta$. Next, conditioned on $x \in \{\xi_i\}_{i \in [u/2]}$, $(\xi_1, \ell_1)$ is assigned high probability $(1 - \varepsilon)$ and the rest of the measure is distributed uniformly over $\{(\xi_i, \ell_i)\}_{i \in [2,u/2]}$. That is

$$\Pr_{\mathcal{D}_\ell}[(\xi_1, \ell_1)] = (1 - \beta)(1 - \varepsilon) \, , \text{ and } \forall j \in [2, u/2]. \ \Pr_{\mathcal{D}_\ell}[(\xi_j, \ell_j)] = \frac{(1 - \beta)\varepsilon}{u/2 - 1} \, .$$

Finally, conditioned on $x \in \{\xi_i\}_{i \in [u/2+1,u]}$, $x$ distributes uniformly over $\{\xi_i\}_{i \in [u/2+1,u]}$, and conditioned on $x = \xi_i$, we have $y = \ell_i$ with probability $\frac{1+\alpha}{2}$. That is

$$\forall j \in [u/2 + 1, u]. \ \Pr_{\mathcal{D}_\ell}[(\xi_j, \ell_j)] = \frac{(1 + \alpha)\beta}{2d} \, , \text{ and } \Pr_{\mathcal{D}_\ell}[(\xi_j, -\ell_j)] = \frac{(1 - \alpha)\beta}{2d} \, .$$

In order to give a lower bound on the out-of-sample error for an arbitrary voting classifier $f$, we define a new random variable that is dominated by $\Pr_{(x,y)\sim\mathcal{D}_\ell}[yf(x) < 0]$, and give a lower bound on that random variable. To this end, fix some $\ell \in \{-1, 1\}^u$ and $f : \mathcal{X} \to \mathbb{R}$, and denote

$$\Psi_1(\ell, f) = \frac{(1 - \varepsilon)\beta}{u/2 - 1} \sum_{i \in [2,u/2]} \mathbb{1}_{\ell_i f(\xi_i) < 0} \quad ; \quad \Psi_2(\ell, f) = \frac{\alpha\beta}{d} \sum_{i \in [u/2+1,u]} \mathbb{1}_{\ell_i f(\xi_i) < 0} \, . \quad (3)$$

When $f, \ell$ are clear from the context we shall simply denote $\Psi_1, \Psi_2$. In this notation, we show the following.

**Claim 4.** $\Pr_{(x,y)\sim\mathcal{D}_\ell}[yf(x) < 0] \geq \frac{\beta(1-\alpha)}{2} + \Psi_1 + \Psi_2$.

While the proof of the claim is deferred to the full version of the paper [GKL$^+$19], we explain why we focus on $\Psi_1 + \Psi_2$, rather than bounding the out-of-sample error directly. The reason lies in the fact that we need a lower bound to hold *with constant probability* over the choice of $\ell$ and $S$ (and in the case of Theorem 1 also the random choices made by the algorithm) and not only *in expectation*. While lower bounding $\mathbb{E}[\Pr_{(x,y)\sim\mathcal{D}_\ell}[yf(x) < 0]]$ is clearly not harder than lower bounding $\mathbb{E}[\Psi_1 + \Psi_2]$, showing that a lower bound holds with some constant probability is slightly more delicate. Our proof uses the fact that with probability 1, $\Psi_1 + \Psi_2$ is not larger than a constant from its expectation, and therefore we can use Markov's inequality to lower bound $\Psi_1 + \Psi_2$ with constant probability.

We next show that there exists a small enough (with respect to $N$) hypothesis set $\hat{\mathcal{H}}$ that is rich enough. That is, with high probability over $\ell \in \{-1, 1\}^u$, there exists a weighted average $f \in C(\hat{\mathcal{H}})$ that attains margin at least $\theta$ over the entire set $\mathcal{X}$. The following claim follows from Lemma 3 and Yao's minimax principle. Its proof is deferred to the full version of the paper [GKL$^+$19].

**Claim 5.** *There exists a hypothesis set $\hat{\mathcal{H}}$ such that $\ln|\hat{\mathcal{H}}| = \Theta(\ln N)$ and*

$$\Pr_{\ell \in_R \{-1,1\}^u}[\exists f \in C(\hat{\mathcal{H}}) : \forall i \in [u]. \ \ell_i f(\xi_i) \geq \theta] \geq 19/20 \, .$$

We next show that there exist some distribution $\mathcal{D} \in \{\mathcal{D}_\ell\}_{\ell \in \{-1,1\}^u}$ and some classifier $\hat{f} \in C(\hat{\mathcal{H}})$ such that for every algorithm $\mathcal{A}$, with constant probability over $S \sim \mathcal{D}_\ell$, $\hat{f}$ has large margins on points in $S$, yet $f_{\mathcal{A},S}$ has large out-of-sample error. To this end fix $\mathcal{A}$ to be a (perhaps randomized) learning algorithm. For every $m$-point sample $S$, recall that $f_{\mathcal{A},S}$ denotes the (random) classifier returned by $\mathcal{A}$ when running on sample $S$.

The main challenge is to show that there exists a labeling $\hat{\ell} \in \{-1, 1\}^u$ such that $C(\hat{\mathcal{H}})$ contains a good voting classifier for $\hat{\ell}$ and, in addition, $f_{\mathcal{A},S}$ has a out-of-sample error with respect to $\mathcal{D}_{\hat{\ell}}$. We will show that if $\alpha$ is small enough, then indeed such a labeling exists. Formally, we show the following.

**Lemma 6.** *If $\alpha \leq \sqrt{\frac{u}{40\beta m}}$, then there exists $\hat{\ell} \in \{-1, 1\}^u$ such that*

1. *There exists $\hat{f} = \hat{f}_{\hat{\ell}} \in C(\hat{\mathcal{H}})$ such that for every $i \in [u]$, $\hat{\ell}_i \hat{f}(\xi_i) \geq \theta$ ; and*

2. *with probability at least $1/25$ over $S \sim \mathcal{D}_{\hat{\ell}}^m$ and the randomness of $\mathcal{A}$ we have*

$$\Psi_1(\hat{\ell}, f_{\mathcal{A},S}) + \Psi_2(\hat{\ell}, f_{\mathcal{A},S}) \geq \frac{(1-\beta)\varepsilon}{24} + \frac{\alpha\beta}{24} \; .$$

The proof of the lemma is quite involved technically, and is therefore also deferred to the full version of the paper [GKL$^+$19]. We will next show that the lemma implies Theorem 1.

*Proof of Theorem 1.* Fix some $\tau \in [0, 49/100]$. Assume first that $\tau \leq \frac{u}{300m}$ , and let $\varepsilon = \frac{u}{10m}$ and $\beta = \alpha = 0$. Let $\hat{\ell}, \hat{f}$ be as in Lemma 6, then for every sample $S \sim \mathcal{D}_{\hat{\ell}}^m$, $\Pr_{(x,y)\sim S}[y\hat{f}(x) < \theta] = 0 \leq \tau$, and moreover with probability at least $1/25$ over $S$ and the randomness of $\mathcal{A}$

$$\Pr_{(x,y)\sim\mathcal{D}_{\hat{\ell}}}[yf_{\mathcal{A},S}(x) < 0] \geq \frac{(1-\beta)\varepsilon}{24} \geq \tau + \Omega\left(\frac{u}{m}\right) = \tau + \Omega\left(\frac{\ln|\hat{\mathcal{H}}|}{m\theta^2} + \sqrt{\frac{\tau\ln|\hat{\mathcal{H}}|}{m\theta^2}}\right) \; .$$

where the first inequality follows from Claim 4 and the second point of Lemma 6, and the last transition is due to the fact that $u = 2\theta^{-2}\ln N = \Theta(\theta^{-2}\lg|\hat{\mathcal{H}}|)$ and $\tau = O(u/m)$.

Otherwise, assume $\tau > \frac{u}{300m}$ , and let $\varepsilon = \frac{u}{10m}$, $\alpha = \sqrt{\frac{u}{2560\tau m}}$ and $\beta = \frac{64\tau}{32-31\alpha}$. Since $\tau \geq \frac{u}{300m}$, then $\alpha \in [0, 1]$. Moreover, if $m > Cu$ for large enough but universal constant $C > 0$, then $32 - 31\alpha \geq 64 \cdot \frac{49}{100} \geq 64\tau$, and hence $\beta \in [0, 1]$. Moreover, since $\alpha \leq 1$ then $\beta \leq 64\tau$, and therefore $\alpha = \sqrt{\frac{u}{2560\tau m}} \leq \sqrt{\frac{u}{40\beta m}}$. Let therefore $\hat{\ell}, \hat{f}$ be a labeling and a classifier in $C(\hat{\mathcal{H}})$ whose existence is guaranteed in Lemma 6. Let $\langle(x_1, y_1), \ldots, (x_m, y_m)\rangle \sim \mathcal{D}_{\hat{\ell}}^m$ be a sample of $m$ points drawn independently according to $\mathcal{D}_{\hat{\ell}}$. For every $j \in [m]$, we have $\mathbb{E}[\mathbb{1}_{y_j\hat{f}(x_j)<\theta}] = \frac{(1-\alpha)\beta}{2}$. Therefore by Chernoff we get that for large enough $N$,

$$\Pr_{S\sim\mathcal{D}_{\hat{\ell}}^m}\left[\Pr_{(x,y)\sim S}\left[y\hat{f}(x) < \theta\right] \geq \tau\right] = \Pr_{S\sim\mathcal{D}_{\hat{\ell}}^m}\left[\frac{1}{m}\sum_{j\in[m]}\mathbb{1}_{y_j\hat{f}(x_j)<\theta} \geq \frac{(1-31\alpha/32)\beta}{2}\right]$$

$$\leq e^{-\Theta(\alpha^2\beta m)} \leq e^{-\Theta(u)} \leq 10^{-3} \; ,$$

where the second-to-last inequality is due to the fact that $\alpha^2\beta m = \frac{u\beta}{2560\tau} = \Omega(u)$, since $\beta \geq 2\tau$. Moreover, from Claim 4 and the second point of Lemma 6 we get that with probability at least $1/25$ over $S$ and $\mathcal{A}$ we have

$$\Pr_{(x,y)\sim\mathcal{D}_{\hat{\ell}}}[yf_{\mathcal{A},S}(x) < 0] \geq \frac{(1-\alpha)\beta}{2} + \frac{\alpha\beta}{32} = \frac{(1-31\alpha/32)\beta}{2} + \frac{\alpha\beta}{64} = \tau + \Omega\left(\sqrt{\frac{\tau u}{m}}\right)$$

$$\geq \tau + \Omega\left(\frac{\ln|\hat{\mathcal{H}}|}{m\theta^2} + \sqrt{\frac{\tau\ln|\hat{\mathcal{H}}|}{m\theta^2}}\right) \; ,$$

where the last transition is due to the fact that $\tau = \Omega(u/m)$. $\qquad\square$

## 4 Existence of a Small Hypotheses Set

This section is devoted to the proof of Lemma 3. That is, we present a distribution $\mu$ over fixed-size hypothesis sets and show that for every fixed labeling $\ell$ with not too many negative labels, with high probability over $\mathcal{H} \sim \mu$, $C(\mathcal{H})$ contains a voting classifier $f$ that attains good margins with respect to $\ell$. In fact, our proof not only shows existence of such a voting classifier, but also presents a procedure for constructing one. The presented algorithm is an adaptation of the AdaBoost algorithm.

More formally, fix some $\theta \in (0, 1/40)$, $\delta \in (0, 1)$ and an integer $d \leq u$. Let $\gamma = 4\theta \in (0, 1/10)$ and let $N = 2\gamma^{-2} \ln d \cdot \ln \frac{\gamma^{-2} \ln d}{\delta} \cdot e^{O(\theta^2 d)}$. We define the distribution $\mu$ via the following procedure, that samples a hypothesis set $\mathcal{H} \sim \mu$. Let $\hat{h}$ be defined by $\hat{h}(x) = 1$ for all $x \in \mathcal{X}$. Sample independently and uniformly at random $N$ hypotheses $h_1, \ldots, h_N$, and define $\mathcal{H} := \{\hat{h}\} \cup \{h_j\}_{j \in [N]}$.

Clearly every $\mathcal{H} \in \mathrm{supp}(\mu)$ satisfies $|\mathcal{H}| = N + 1$. We therefore turn to prove the second property. To this end, let $k = \gamma^{-2} \ln d$. In order to show existence of a voting classifier, we conceptually change the procedure defining $\mu$, and think of the random hypotheses as being sampled in $k$ equally sized "batches", each of size $N/k$, and adding $\hat{h}$ to each of them. Denote the batches by $\mathcal{H}_1, \mathcal{H}_2, \ldots, \mathcal{H}_k$. We consider next the following procedure to construct a voting classifier $f \in C(\mathcal{H})$ given $\mathcal{H} \sim \mu$. We will use the main ideas from the AdaBoost algorithm. Recall that AdaBoost creates a voting classifier using a sample $S = ((x_1, y_1), \ldots, (x_u, y_u))$ in iterations. Staring with $f_0 = 0$, in iteration $j$, it computes a new voting classifier $f_j = f_{j-1} + \alpha_j h_j$ for some hypothesis $h_j \in \mathcal{H}$ and weight $\alpha_j$. The heart of the algorithm lies in choosing $h_j$. In each iteration, AdaBoost computes a distribution $D_j$ over $S$ and chooses a hypothesis $h_j$ minimizing the empirical error probability $\varepsilon_j = \mathrm{Pr}_{i \sim D_j}[h_j(x_i) \neq y_i]$ with respect to $D_j$ and then reweighs the sample points to construct $D_{j+1}$. The weight it then assigns to $h_j$ is $\alpha_j = (1/2) \ln((1 - \varepsilon_j)/\varepsilon_j)$ The first distribution $D_1$ is the uniform distribution.

We alter the above slightly assigning uniform weights on the hypotheses, and setting $\alpha_j = \frac{1}{2} \ln \frac{1+2\gamma}{1-2\gamma}$ for all iterations $j$. The algorithm is formally described as Algorithm 1.

---

**Input:** $(\mathcal{H}_1, \ldots, \mathcal{H}_k) \sim \mu$
**Output:** $f \in C\left(\bigcup_{j \in [k]} \mathcal{H}_j\right)$
1: let $\alpha = \frac{1}{2} \ln \frac{1+2\gamma}{1-2\gamma}$
2: let $f(x) = 0$ for all $x \in \mathcal{X}$
3: let $D_1(i) = \frac{1}{u}$ for all $i \in [u]$.
4: **for** $j = 1$ to $k$ **do**
5:     Find a hypothesis $h_j \in \mathcal{H}_j$ satisfying $\sum_{i \in [u]} D_j(i) \mathbb{1}_{y_i \neq h_j(x_i)} \leq \frac{1}{2} - \gamma$.
    If there is no such hypothesis, **return** *fail*.
6:     $f_j \leftarrow f_{j-1} + h_j$.
7:     $Z_j \leftarrow \sum_{i \in [u]} D_j(i) \exp(-\alpha y_i h_j(x_i))$.
8:     for every $i \in [u]$ let $D_{j+1}(i) = \frac{1}{Z_j} D_j(i) \exp(-\alpha y_i h_j(x_i))$.
9: **return** $\frac{1}{k} f_k$.

**Algorithm 1:** Construct a Voting Classifier

---

First note that if $f$ is the classifier returned by the algorithm, then clearly $f = \frac{1}{k} \sum_{j \in [k]} h_j \in C(\mathcal{H})$ is a voting classifier. The following claim implies Lemma 3. Its proof is quite technical, and deferred to the full version of the paper [GKL$^+$19].

**Claim 7.** *With probability at least $1 - \delta$ Algorithm 1 does not fail, and moreover, in that case, for every $i \in [y]$, $y_i f(x_u) \geq \theta$.*

## 5 Conclusions

In this work, we showed almost tight margin-based generalization lower bounds for voting classifiers. These new bounds essentially complete the theory of generalization for voting classifers based on margins alone. Closing the remaining gap between the upper and lower bounds is an intriguing open problem and we hope our techniques might inspire further improvements. Our results come in the form of two theorems, one showing generalization lower bounds for *any* algorithm producing a voting classifier, and a slightly stronger lower bound showing the *existence* of a voting classifier with poor generalization. This raises the important question of whether specific boosting algorithms can produce voting classifiers that avoid the $\lg m$ factor in the second lower bound via a careful analysis tailored to the algorithm. As a final important direction for future work, we suggest investigating whether natural parameters other than margins may be used to better explain the practical generalization error of voting classifiers. At least, we now have an almost tight understanding, if no further parameters are taken into consideration.

**Acknowledgments**

This work was supported by a Villum Young Investigator Grant and an AUFF Starting Grant.

Jelani Nelson is supported by NSF CAREER award CCF-1350670, NSF grant IIS-1447471, ONR grant N00014-18-1-2562, ONR DORECG award N00014-17-1-2127, an Alfred P. Sloan Research Fellowship, and a Google Faculty Research Award

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
