[Supplementary Material · marginLowerBoundsNeurIPS2019Full.pdf]

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

Note that Lemma 3 speaks of a distribution over hypothesis sets. When using Lemma 3 in our proofs, we will invoke Yao's principle to conclude the existence of a suitable fixed hypothesis set $\mathcal{H}$.

**Existential Lower Bound.** Our proof of Theorem 2 uses many of the same ideas as the proof of Theorem 1. The difference between the generalization lower bound (second point) in Theorem 1 and 2 is an $\ln m$ factor in the first term inside the $\Omega(\cdot)$ notation. That is, Theorem 2 has an $\Omega(\frac{\ln |\mathcal{H}| \ln m}{\theta^2 m})$ where Theorem 1 has an $\Omega(\frac{\ln |\mathcal{H}|}{\theta^2 m})$. This term originated from having $\ln |\mathcal{H}|/\theta^2$ points with a probability mass of $1/10m$ in $\mathcal{D}_\ell$ and one point having the remaining probability mass. In the proof of Theorem 2, we first exploit that we are proving an existential lower bound by assigning all points the same label 1. That is, our hard distribution $\mathcal{D}$ assigns all points the label 1 (ignoring the second half of the distribution with the random and slightly biased labels). Since we are not proving a lower bound for every algorithm, this will not cause problems. We then change $|\mathcal{X}|$ to about $m/\ln m$ and assign each point the same probability mass $\ln m/m$ in distribution $\mathcal{D}$. The key observation is that on a random sample $S$ of $m$ points, by a coupon-collector argument, there will still be $m^{\Omega(1)}$ points from $\mathcal{X}$ that were not sampled. From Lemma 3, we can now find a voting classifier $f$, such that $\mathrm{sign}(f(x))$ is 1 on all points in $x \in S$, and $-1$ on a set of $d = \ln |\mathcal{H}|/\theta^2$ points in $\mathcal{X} \setminus S$. This means that $f$ has out-of-sample error $\Omega(d \ln m/m) = \Omega(\frac{\ln |\mathcal{H}| \ln m}{\theta^2 m})$ under distribution $\mathcal{D}$ and obtains a margin of $\theta$ on all points in the sample $S$.

As in the proof Theorem 1, we can combine the above distribution $\mathcal{D}$ with the ideas of Anthony and Bartlett to add the terms depending on $\tau$ to the lower bound.

## 3 Margin-Based Generalization Lower Bounds

In this section we prove Theorems 1 and 2 assuming Lemma 3, whose proof is deferred to Section 4, and we start by describing the outlines of the proofs. To this end fix some integer $N$, and fix $\theta \in (1/N, 1/40)$. Let $u$ be an integer, and let $\mathcal{X} = \{\xi_1, \ldots, \xi_u\}$ be some set with $u$ elements. With every $\ell \in \{-1, 1\}^u$ we associate a distribution $\mathcal{D}_\ell$ over $\mathcal{X} \times \{-1, 1\}$, and show that with some constant probability over a random choice of $\ell$, a voting classifier of interest has a high generalization probability with respect to $\mathcal{D}_\ell$. By a voting classifier of interest we mean one constructed by a learning algorithm in the proof of Theorem 1 and an adversarial classifier in the proof of Theorem 2. We additionally show existence of a hypothesis set $\hat{\mathcal{H}}$ such that with very high (constant) probability over a random choice of $\ell \in \{-1, 1\}^u$, $C(\hat{\mathcal{H}})$ contains a voting classifier that attains high margins with $\ell$ over the entire set $\mathcal{X}$. Finally, we conclude that with positive probability over a random choice of $\ell \in \{-1, 1\}^u$ both properties are satisfied, and therefore there exists at least one labeling $\ell$ that satisfies both properties.

274 We start by constructing the family $\{\mathcal{D}_\ell\}_{\ell \in \{-1,1\}^u}$ of distributions over $\mathcal{X} \times \{-1,1\}$. To this end,
275 let $d \le u$ be some constant to be fixed later, and let $\ell \in \{-1,1\}^u$. We define $\mathcal{D}_\ell$ separately for the
276 first $u - d$ points and the last $d$ points of $\mathcal{X}$. Intuitively, every point in $\{\xi_i\}_{i\in[u-d]}$ has a fixed label
277 determined by $\ell$, however all points but one have a very small probability of being sampled according
278 to $\mathcal{D}_\ell$. Every point in $\{\xi_i\}_{i\in[u-d,u]}$, on the other hand, has an equal probability of being sampled,
279 however its label is not fixed by $\ell$ rather than slightly biased towards $\ell$. Formally, let $\alpha, \beta, \varepsilon \in [0,1]$
280 be constants to fixed later. We construct $\mathcal{D}_\ell$ using the ideas described earlier in Section 2, by
281 sewing them together over two parts of the set $\mathcal{X}$. We assign probability $1 - \beta$ to $\{\xi_i\}_{i\in[u-d]}$ and
282 $\beta$ to $\{\xi_i\}_{i\in[u-d+1,u]}$. That is, for $(x,y) \sim \mathcal{D}_\ell$, the probability that $x \in \{\xi_i\}_{i\in[u-d]}$ is $1 - \beta$. Next,
283 conditioned on $x \in \{\xi_i\}_{i\in[u-d]}$, $(\xi_1, \ell_1)$ is assigned high probability $(1 - \varepsilon)$ and the rest of the
284 measure is distributed uniformly over $\{(\xi_i, \ell_i)\}_{i\in[2,u-d]}$. That is

$$\Pr_{\mathcal{D}_\ell}[(\xi_1, \ell_1)] = (1 - \beta)(1 - \varepsilon) , \ and \ \forall j \in [2, u-d]. \ \Pr_{\mathcal{D}_\ell}[(\xi_j, \ell_j)] = \frac{(1-\beta)\varepsilon}{u-d-1} .$$

285 Finally, conditioned on $x \in \{\xi_i\}_{i\in[u-d+1,u]}$, $x$ distributes uniformly over $\{\xi_i\}_{i\in[u-d+1,u]}$, and
286 conditioned on $x = \xi_i$, we have $y = \ell_i$ with probability $\frac{1+\alpha}{2}$. That is

$$\forall j \in [u-d+1, u]. \ \Pr_{\mathcal{D}_\ell}[(\xi_j, \ell_j)] = \frac{(1+\alpha)\beta}{2d} , \ and \ \Pr_{\mathcal{D}_\ell}[(\xi_j, -\ell_j)] = \frac{(1-\alpha)\beta}{2d} .$$

287 In order to give a lower bound on the generalization error for some classifier $f$ of interest, we define
288 new random variables such that their sum is upper bounded by $\Pr_{(x,y)\sim\mathcal{D}_\ell}[yf(x) < 0]$, and give a
289 lower bound on that sum. To this end, for every $\ell \in \{-1,1\}^u$ and $f : \mathcal{X} \to \mathbb{R}$, denote

$$\Psi_1(\ell, f) = \frac{(1-\varepsilon)\beta}{u-d-1} \sum_{i\in[2,u-d]} \mathbb{1}_{\ell_i f(\xi_i)<0} \quad ; \quad \Psi_2(\ell, f) = \frac{\alpha\beta}{d} \sum_{i\in[u-d+1,u]} \mathbb{1}_{\ell_i f(\xi_i)<0} . \quad (3)$$

290 When $f, \ell$ are clear from the context we shall simply denote $\Psi_1, \Psi_2$. We show next that indeed
291 proving a lower bound on $\Psi_1 + \Psi_2$ implies a lower bound on the generalization error.

292 **Claim 4.** *For every $\ell, f$ we have* $\Pr_{(x,y)\sim\mathcal{D}_\ell}[yf(x) < 0] \ge \frac{\beta(1-\alpha)}{2} + \Psi_1 + \Psi_2$.

293 Before getting proving the claim, we explain why focusing on $\Psi_1 + \Psi_2$, rather than bounding the
294 generalization error directly is essential for the proof. The reason lies in the fact that we need a lower
295 bound to hold *with constant probability* over the choice of $\ell$ and $S$ (and in the case of Theorem 1
296 also the random choices made by the algorithm) and not only *in expectation*. While lower bounding
297 $\mathbb{E}[\Pr_{(x,y)\sim\mathcal{D}_\ell}[yf(x) < 0]]$ is clearly not harder than lower bounding $\mathbb{E}[\Psi_1 + \Psi_2]$, showing that a
298 lower bound holds with some constant probability is slightly more delicate. Our proof uses the fact
299 that with probability 1, $\Psi_1 + \Psi_2$ is not larger than a constant from its expectation, and therefore we
300 can use Markov's inequality to lower bound $\Psi_1 + \Psi_2$ with constant probability. We next turn to
301 prove the claim.

302 *Proof.* We first observe that

$$\Pr_{(x,y)\sim\mathcal{D}_\ell}[yf(x) < 0] = \mathbb{E}_{(x,y)\sim\mathcal{D}_\ell}[\mathbb{1}_{yf(x)<0}]$$

$$= \sum_{i\in[u-d],y\in\{-1,1\}} \mathbb{1}_{yf(\xi_i)<0} \Pr_{\mathcal{D}_\ell}[(\xi_i, y)] + \sum_{i\in[u-d+1,u],y\in\{-1,1\}} \mathbb{1}_{yf(\xi_i)<0} \Pr_{\mathcal{D}_\ell}[(\xi_i, y)] \quad (4)$$

303 For every $i \in [u-d]$ and $y \in \{-1,1\}$, if $y \ne \ell_i$ then $\Pr_{\mathcal{D}_y}[(\xi_j, y)] = 0$. Moreover, if $i \ge 2$ and
304 $y = \ell_i$ then $\Pr_{\mathcal{D}_y}[(\xi_i, y)] = \frac{(1-\beta)\varepsilon}{u-d-1}$. Therefore

$$\sum_{j\in[u-d],y\in\{-1,1\}} \mathbb{1}_{yf(\xi_j)<0} \Pr_{\mathcal{D}_y}[(\xi_j, y)] \ge \frac{(1-\beta)\varepsilon}{u-d-1} \sum_{j\in[2,u-d]} \mathbb{1}_{yf(\xi_j)<0} = \Psi_1 . \quad (5)$$

305 Next, for every $i \in [u-d+1, u]$ we have that

$$\sum_{y\in\{-1,1\}} \mathbb{1}_{yf(\xi_i)<0} \Pr_{\mathcal{D}_\ell}[(\xi_i, y)] = \mathbb{1}_{\ell_i f(\xi_i)<0} \Pr_{\mathcal{D}_\ell}[(\xi_i, \ell_i)] + \mathbb{1}_{\ell_i f(\xi_i)>0} \Pr_{\mathcal{D}_\ell}[(\xi_i, -\ell_i)]$$

$$= \frac{(1-\alpha)\beta}{2d} + \mathbb{1}_{\ell_i f(\xi_i)<0} \frac{\alpha\beta}{d} ,$$

and therefore

$$\sum_{i\in[u-d+1,u],y\in\{-1,1\}} \mathbb{1}_{yf(\xi_i)<0} \Pr_{\mathcal{D}_\ell}[(\xi_i,y)] = \frac{(1-\alpha)\beta}{2} + \frac{\alpha\beta}{d}\sum_{i\in[u-d+1,u]} \mathbb{1}_{\ell_i f(\xi_i)<0} \ . \quad (6)$$

Plugging (5) and (6) into (4) we conclude the claim. □

To prove existence of a "rich" yet small enough hypothesis set $\hat{\mathcal{H}}$ we apply Lemma 3 together with Yao's minimax principle. In order to ensure that the hypothesis sets constructed using Lemma 3 is small enough, and specifically has size $N^{O(1)}$, we need to focus our attention on sparse labelings $\ell \in \{-1,1\}^u$ only. That is, the labelings cannot contain more than $\Theta\left(\frac{\ln N}{\theta^2}\right)$. To this end we will focus on $2d$-sparse vectors, and more specifically, a designated set of $2d$-sparse labelings. More formally, we define a set of labelings of interest $\mathcal{L}(u,d)$ as the set of all labelings $\ell \in \{-1,1\}^u$ such that the restriction to the first $u-d$ entries is $d$-sparse. That is

$$\mathcal{L}(u,d) := \{\ell \in \{-1,1\}^u : |\{i \in [u-d] : \ell_i = -1\}| \leq d\} \ . \quad (7)$$

We next show that there exists a small enough (with respect to $N$) hypothesis set $\hat{\mathcal{H}}$ that is rich enough. That is, with high probability over $\ell \in \mathcal{L}(u,d)$, there exists a voting classifier $f \in C(\hat{\mathcal{H}})$ that attains high minimum margin with $\ell$ over the entire set $\mathcal{X}$. Note that the following result, similarly to Lemma 3 does not depend on the size of $\mathcal{X}$, but only on the sparsity of the labelings in question.

**Claim 5.** *If $d \leq \frac{\ln N}{\theta^2}$ then there exists a hypothesis set $\hat{\mathcal{H}}$ such that $\ln|\hat{\mathcal{H}}| = \Theta(\ln N)$ and*

$$\Pr_{\ell\in_R\mathcal{L}(u,d)}[\exists f \in \mathcal{C}(\hat{\mathcal{H}}) : \forall i \in [u]. \ \ell_i f(\xi_i) \geq \theta] \geq 1 - 1/N \ .$$

*Proof.* Let $\mu = \mu(u,d,\theta,1/N)$, be the distribution whose existence is guaranteed in Lemma 3. Then for every labeling $\ell \in \mathcal{L}(u,d)$, with probability at least $99/100$ over $\mathcal{H} \sim \mu$, there exists a voting classifier $f \in C(\mathcal{H})$ that has minimal margin of $\theta$. That is, for every $i \in [u]$, $\ell_i f(\xi_i) \geq \theta$. By Yao's minimax principle, there exists a hypothesis set $\hat{\mathcal{H}} \in \text{supp}(\mu)$ such that

$$\Pr_{\ell\in_R\mathcal{L}(u,d)}[\exists f \in \mathcal{C}(\hat{\mathcal{H}}) : \forall i \in [u]. \ \ell_i f(x_i) \geq \theta] \geq 1 - 1/N \ .$$

Moreover, since $\hat{\mathcal{H}} \in \text{supp}(\mu)$, then $|\hat{\mathcal{H}}| = \Theta\left(\theta^{-2}\ln d \cdot \ln(N\theta^{-2}\ln d) \cdot e^{\Theta(\theta^2 d)}\right)$. Since $\theta \geq 1/N$ and since $d = \frac{\ln N}{\theta^2}$ and thus $e^{\theta^2 d} = N$ we get that there exists some univeral constant $C > 0$ such that $|\hat{\mathcal{H}}| = \Theta(N^C)$, and thus $\ln|\hat{\mathcal{H}}| = \Theta(\ln N)$. □

### 3.1 Proof Algorithmic Lower Bound

This section is devoted to the proof of Theorem 1. That is, we show that for every algorithm $\mathcal{A}$, there exist some distribution $\mathcal{D} \in \{\mathcal{D}_\ell\}_{\ell\in\{-1,1\}^u}$ and some classifier $\hat{f} \in C(\hat{\mathcal{H}})$ such that with constant probability over $S \sim \mathcal{D}^m$, $\hat{f}$ has large margins on points in $S$, yet $f_{\mathcal{A},S}$ has large generalization error. To this end we now fix $u$ to be $\frac{2\ln N}{\theta^2}$ and $d = \frac{u}{2} = \frac{\ln N}{\theta^2}$. For these values of $u,d$ we get that $\mathcal{L}(u,d)$ is, in fact, the set of all possible labelings, i.e. $\mathcal{L}(u,d) = \{-1,1\}^u$. Next, fix $\mathcal{A}$ be a (perhaps randomized) learning algorithm. For every $m$-point sample $S$ and recall that $f_{\mathcal{A},S}$ denotes the classifier returned by $\mathcal{A}$ when running on sample $S$.

The main challenge is to show that there exists a labeling $\hat{\ell} \in \{-1,1\}^u$ such that $C(\hat{\mathcal{H}})$ contains a good voting classifier for $\hat{\ell}$ and, in addition, $f_{\mathcal{A},S}$ has a large generalization error with respect to $\mathcal{D}_{\hat{\ell}}$. We will show that if $\alpha$ is small enough, then indeed such a labeling exists. Formally, we show the following.

**Lemma 6.** *If $\alpha \leq \sqrt{\frac{u}{40\beta m}}$, then there exists $\hat{\ell} \in \{-1,1\}^u$ such that*

1. *There exists $\hat{f} = \hat{f}_{\hat{\ell}} \in \mathcal{C}(\hat{\mathcal{H}})$ such that for every $i \in [u]$, $\hat{\ell}_i \hat{f}(\xi_i) \geq \theta$ ; and*

2. *with probability at least $1/25$ over $S \sim \mathcal{D}_{\hat{\ell}}^m$ and the randomness of $\mathcal{A}$ we have*

$$\Pr_{(x,y)\sim\mathcal{D}_{\hat{\ell}}}[yf_{\mathcal{A},S}(x) < 0] \geq \frac{(1-\alpha)\beta}{2} + \frac{(1-\beta)\varepsilon}{24} + \frac{\alpha\beta}{24} \ .$$

340    Before proving the lemma, we first show how it implies Theorem 1

341    *Proof of Theorem 1.* Fix some $\tau \in [0, 49/100]$. Assume first that $\tau \leq \frac{u}{300m}$ , and let $\varepsilon = \frac{u}{10m}$ and
342    $\beta = \alpha = 0$. Let $\hat{\ell}, \hat{f}$ be as in Lemma 6, then for every sample $S \sim \mathcal{D}_{\hat{\ell}}^m$, $\Pr_{(x,y)\sim S}[y\hat{f}(x) < \theta] =$
343    $0 \leq \tau$, and moreover with probability at least $1/25$ over $S$ and the randomness of $\mathcal{A}$

$$\Pr_{(x,y)\sim\mathcal{D}_{\hat{\ell}}}[y f_{\mathcal{A},S}(x) < 0] \geq \frac{(1-\beta)\varepsilon}{24} \geq \tau + \Omega\left(\frac{u}{m}\right) = \tau + \Omega\left(\frac{\ln|\hat{\mathcal{H}}|}{m\theta^2} + \sqrt{\frac{\tau \ln|\hat{\mathcal{H}}|}{m\theta^2}}\right) .$$

344    where the last transition is due to the fact that $u = 2\theta^{-2}\ln N = \Theta(\theta^{-2}\lg|\hat{\mathcal{H}}|)$ and $\tau = O(u/m)$.

345    Otherwise, assume $\tau > \frac{u}{300m}$ , and let $\varepsilon = \frac{u}{10m}, \alpha = \sqrt{\frac{u}{2560\tau m}}$ and $\beta = \frac{64\tau}{32-31\alpha}$. Since $\tau \geq \frac{u}{300m}$,
346    then $\alpha \in [0, 1]$. Moreover, if $m > Cu$ for large enough but universal constant $C > 0$, then
347    $32 - 31\alpha \geq 64 \cdot \frac{49}{100} \geq 64\tau$, and hence $\beta \in [0, 1]$. Moreover, since $\alpha \leq 1$ then $\beta \leq 64\tau$, and
348    therefore $\alpha = \sqrt{\frac{u}{2560\tau m}} \leq \sqrt{\frac{u}{40\beta m}}$. Let therefore $\hat{\ell}, \hat{f}$ be a labeling and a classifier in $C(\hat{\mathcal{H}})$
349    whose existence is guaranteed in Lemma 6. Let $\langle(x_1, y_1), \ldots, (x_m, y_m)\rangle \sim \mathcal{D}_{\hat{y}}^m$ be a sample of $m$
350    points drawn independently according to $\mathcal{D}_{\hat{\ell}}$. For every $j \in [m]$, we have $\mathbb{E}[\mathbb{1}_{y_j\hat{f}(x_j)<\theta}] = \frac{(1-\alpha)\beta}{2}$.
351    Therefore by Chernoff we get that for large enough $N$,

$$\Pr_{S\sim\mathcal{D}_{\hat{\ell}}^m}\left[\Pr_{(x,y)\sim S}\left[y\hat{f}(x) < \theta\right] \geq \tau\right] = \Pr_{S\sim\mathcal{D}_{\hat{\ell}}^m}\left[\frac{1}{m}\sum_{j\in[m]}\mathbb{1}_{\hat{y}_j\hat{f}(x_j)<\theta} \geq \frac{(1-31\alpha/32)\beta}{2}\right]$$
$$\leq e^{-\Theta(\alpha^2\beta m)} \leq e^{-\Theta(u)} \leq 10^{-3} ,$$

352    where the inequality before last is due to the fact that $\alpha^2\beta m = \frac{u\beta}{2560\tau} = \Omega(u)$, since $\beta \geq 2\tau$.
353    Moreover, by Lemma 6 we get that with probability at least $1/25$ over $S$ and $\mathcal{A}$ we get that

$$\Pr_{(x,y)\sim\mathcal{D}_{\hat{\ell}}}[y f_{\mathcal{A},S}(x) < 0] \geq \frac{(1-\alpha)\beta}{2} + \frac{\alpha\beta}{32} = \frac{(1-31\alpha/32)\beta}{2} + \frac{\alpha\beta}{64} = \tau + \Omega\left(\sqrt{\frac{\tau u}{m}}\right)$$
$$\geq \tau + \Omega\left(\frac{\ln|\hat{\mathcal{H}}|}{m\theta^2} + \sqrt{\frac{\tau \ln|\hat{\mathcal{H}}|}{m\theta^2}}\right) ,$$

354    where the last transition is due to the fact that $\tau = \Omega(u/m)$. This completes the proof of Theorem 1.
355    $\qquad\qquad\qquad\qquad\qquad\qquad\qquad\qquad\qquad\qquad\qquad\qquad\qquad\qquad\qquad\qquad\qquad\qquad\qquad$ $\square$

356    For the rest of the section we therefore prove Lemma 6. We start by lower bounding the expected
357    value of $\Psi_1 + \Psi_2$, where the expectation is over the choice of labeling $\ell \in \{-1, 1\}^u$, $S \sim \mathcal{D}_\ell^m$
358    and the random choices made by $\mathcal{A}$. Intuitively, as points in $\{\xi_2, \ldots, \xi_u\}$ are sampled with very
359    small probability, it is very likely that the sample $S$ does not contain many of them, and therefore
360    the algorithm cannot do better than randomly guessing many of the labels. Moreover, if $\alpha$ is small
361    enough, and $S$ does not sample a point in $\{\xi_{u/2+1}, \ldots, \xi_u\}$ enough times, there is a larger probability
362    that $\mathcal{A}$ does not determine the bias correctly.

363    **Claim 7.** *If $\alpha \leq \sqrt{\frac{u}{40\beta m}}$, then $\mathbb{E}_{\ell\in\{-1,1\}^u}\left[\mathbb{E}_{\mathcal{A},S}\left[\Psi_1(\ell, f_{\mathcal{A},S}) + \Psi_2(\ell, f_{\mathcal{A},S})\right]\right] \geq \frac{(1-\beta)\varepsilon}{6} + \frac{\alpha\beta}{6}$.*

364    *Proof.* To lower bound the expectation, we lower bound the expectations of $\Psi_1$ and $\Psi_2$ separately.
365    For every $i \in [2, u-d] \setminus \{1\}$, if $\xi_i \notin S$ then $\ell_i$ and $f_{\mathcal{A},S}(\xi_i)$ are independent, and therefore
366    $\mathbb{E}_\ell[\mathbb{1}_{\ell_i f_{\mathcal{A},S}(\xi_i)<0}] = \frac{1}{2}$. Let $\mathcal{S}$ be the set of all samples for which $|S \cap \{\xi_2, \ldots, \xi_{u-d}\}| \leq \frac{u-d-1}{2}$,
367    then for every $S \in \mathcal{S}$,

$$\mathbb{E}_\ell\left[\sum_{i\in[2,u-d-1]}\mathbb{1}_{\ell_i f_{\mathcal{A},S}(\xi_i)<0}\right] \geq \frac{u-d-1-|S\cap\{\xi_2,\ldots,\xi_{u-d}\}|}{2} \geq \frac{u-d-1}{4} ,$$

368 As this holds for every $S \in \mathcal{S}$, we conclude that

$$\mathbb{E}_{\mathcal{A},S}\left[\mathbb{E}_\ell\left[\Psi_1(\ell, f_{\mathcal{A},S})\right] \mid S \in \mathcal{S}\right] \geq \frac{(1-\beta)\varepsilon}{u-d-1} \cdot \frac{u-d-1}{4} = \frac{(1-\beta)\varepsilon}{4} .$$

369 Next, for large enough $N$ a Chernoff bound gives $\Pr_{S \sim \mathcal{D}^m}[\mathcal{S}] \geq 1 - e^{-\Theta(u)} \geq 2/3$, and therefore
370 $\mathbb{E}_{\mathcal{A},S}\left[\mathbb{E}_\ell\left[\Psi_1(\ell, f_{\mathcal{A},S})\right]\right] \geq \frac{(1-\beta)\varepsilon}{6}$, and by Fubini's theorem $\mathbb{E}_\ell\left[\mathbb{E}_{\mathcal{A},S}[\Psi_1(\ell, f_{\mathcal{A},S})]\right] \geq \frac{(1-\beta)\varepsilon}{6}$.

371 Next, let $i \in [u-d+1, u]$. Denote by $\sigma_i \in [m]$ the number of times $\xi_i$ was sampled into $S$. Then

$$\mathbb{E}_\ell\left[\mathbb{E}_{\mathcal{A},S}\left[\mathbb{1}_{\ell_i f_{\mathcal{A},S}(\xi_i)<0}\right]\right] = \sum_{n=0}^m \mathbb{E}_\ell\left[\mathbb{E}_{\mathcal{A},S}\left[\mathbb{1}_{\ell_i f_{\mathcal{A},S}(\xi_i)<0}\big| \sigma_i = n\right]\right] \cdot \Pr[\sigma_i = n] \qquad (8)$$

372 For every $x > 0$ and $y \in (0,1)$, let $\Phi(x,y) = \frac{1}{4}\left(1 - \sqrt{1 - \exp\left(\frac{-xy^2}{1-y^2}\right)}\right)$, then a result by
373 Anthony and Bartlett [AB09, Lemma 5.1] shows that

$$\mathbb{E}_\ell\left[\mathbb{E}_{\mathcal{A},S}\left[\mathbb{1}_{\ell_i f_{\mathcal{A},S}(\xi_i)<0}\big| \sigma_i = n\right]\right] \geq \Phi(n+2, \alpha)$$

374 Plugging this into (8), by the convexity of $\Phi(\cdot, \alpha)$ and Jensen's inequality we get that

$$\mathbb{E}_\ell\left[\mathbb{E}_{\mathcal{A},S}\left[\mathbb{1}_{\ell_i f_{\mathcal{A},S}(\xi_i)<0}\right]\right] \geq \sum_{n=0}^m \Phi(n+2, \alpha) \cdot \Pr[\sigma_i = n] \geq \Phi(\mathbb{E}[\sigma_i] + 2, \alpha) .$$

375 Since $\mathbb{E}[\sigma_i] = \frac{2\beta m}{u}$, and Since $\Phi(\cdot, \alpha)$ is monotonically decreasing we get that

$$\mathbb{E}_\ell\left[\mathbb{E}_{\mathcal{A},S}\left[\mathbb{1}_{\ell_i f_{\mathcal{A},S}(\xi_i)<0}\right]\right] \geq \Phi\left(\frac{4\beta m}{u}, \alpha\right) .$$

376 Summing over all $i \in [u-d+1, u]$ we get that $\mathbb{E}_\ell\left[\mathbb{E}_{\mathcal{A},S}[\Psi_2(\ell, f_{\mathcal{A},S})]\right] \geq \alpha\beta\Phi\left(\frac{4\beta m}{u}, \alpha\right)$. The
377 claim then follows from the fact that for every $\alpha \leq \sqrt{\frac{u}{40\beta m}}$ we have $\Phi(\frac{8\beta m}{u}, \alpha) \geq \frac{1}{6}$. $\qquad\square$

378 We next show that for small values of $\alpha$, a large fraction of labelings $\ell \in \{-1,1\}^u$ satisfy that
379 $\Psi_1 + \Psi_2$ is large with some positive constant probability over the random choices of $\mathcal{A}$ and the choice
380 of $S \in \mathcal{S}$.

381 **Claim 8.** *If $\alpha \leq \sqrt{\frac{u}{40\beta m}}$, then with probability at least $1/11$ over the choice of $\ell \in \{-1,1\}^u$ we*
382 *have*

$$\Pr_{\mathcal{A},S}\left[\Psi_1(\ell, f_{\mathcal{A},S}) + \Psi_2(\ell, f_{\mathcal{A},S}) \geq \frac{(1-\beta)\varepsilon}{24} + \frac{\alpha\beta}{24}\right] \geq \frac{1}{25} .$$

383 *Proof.* First note that by substituting every indicator in (3) with 1 we get that with probability 1 over
384 all samples $S$, labelings $\ell$ and random choices of $\mathcal{A}$ we have

$$\Psi_1 + \Psi_2 \leq (1-\beta)\varepsilon + \alpha\beta , \qquad (9)$$

385 and therefore $\Pr_\ell\left[\mathbb{E}_{\mathcal{A},S}[\Psi_1 + \Psi_2] \leq (1-\beta)\varepsilon + \alpha\beta\right] = 1$. Furthermore, for every $\alpha \leq \sqrt{\frac{u}{40\beta m}}$ we
386 get from Claim 7 that $\mathbb{E}_\ell\left[\mathbb{E}_{\mathcal{A},S}[\Psi_1 + \Psi_2]\right] \geq \frac{1}{6}\left((1-\beta)\varepsilon + \alpha\beta\right)$. Denote $X = \mathbb{E}_{\mathcal{A},S}[\Psi_1 + \Psi_2]$ and
387 $a = (1-\beta)\varepsilon + \alpha\beta$. In these notations we have that (9) states that $\Pr_\ell[X \leq a] = 1$, and Claim 7
388 states that $\mathbb{E}_\ell[X] \geq a/6$. Therefore $a - X$ is a non-negative random variable, and from Markov's
389 inequality we get that

$$\Pr_\ell[X \leq a/12] = \Pr_\ell[a - X \geq 11a/12] \leq \Pr_\ell[a - X \geq 1.1\mathbb{E}[a - X]] \leq 10/11$$

390 and therefore $\Pr_\ell[\mathbb{E}_{\mathcal{A},S}[\Psi_1 + \Psi_2] \geq \frac{1}{12}((1-\beta)\varepsilon + \alpha\beta)] \geq 1/11$.

391 Next, fix some $\ell \in \{-1,1\}^u$ for which $\mathbb{E}_{\mathcal{A},S}[\Psi_1 + \Psi_2] \geq \frac{1}{12}((1-\beta)\varepsilon + \alpha\beta)$. Once again, as
392 $\Pr_{\mathcal{A},S}[\Psi_1 + \Psi_2 \leq 12\mathbb{E}_{\mathcal{A},S}[\Psi_1 + \Psi_2]] = 1$ we get from Markov's inequality that with probability at
393 least $1/25$ we have

$$\Pr_{\mathcal{A},S}\left[\Psi_1 + \Psi_2 \geq \frac{(1-\varepsilon)\beta}{24} + \frac{\alpha\beta}{24}\right] \geq \Pr_{\mathcal{A},S}\left[\Psi_1 + \Psi_2 \geq \frac{1}{2}\mathbb{E}_{\mathcal{A},S}[\Psi_1 + \Psi_2]\right] \geq \frac{1}{25} .$$

394 $\qquad\qquad\qquad\qquad\qquad\qquad\qquad\qquad\qquad\qquad\qquad\qquad\qquad\qquad\qquad\qquad\qquad\qquad\square$

To finish the proof of Lemma 6, observe that from Claims 5 and 8 we get that with positive probability over $\ell \in \{-1, 1\}$ there exists a voting classifier $f \in C(\hat{\mathcal{H}})$ such that for every $i \in [u]$, $\ell_i f(x_i) \geq \theta$ and in addition $\Pr_{\mathcal{A},S} \left[ \Psi_1 + \Psi_2 \geq \frac{(1-\varepsilon)\beta}{24} + \frac{\alpha\beta}{24} \right] \geq \frac{1}{25}$. As this occurs with positve probability, we conclude that there exists some labeling $\hat{\ell} \in \{-1, 1\}^u$ satisfying both properties. Since for every set of random choices of $\mathcal{A}$, and every $S \sim \mathcal{D}_{\hat{\ell}}^m$, Claim 4 guarantees that

$$\Pr_{(x,y)\sim\mathcal{D}_{\hat{\ell}}} [y f_{\mathcal{A},S}(x)] \geq \frac{(1-\alpha)\beta}{2} + \Psi_1(\hat{\ell}, f_{\mathcal{A},S}) + \Psi_2(\hat{\ell}, f_{\mathcal{A},S}) \,,$$

this concludes the proof of Lemma 6, and thus the proof of Theorem 1 is now complete.

## 3.2   Proof of Existential Lower Bound

This section is devoted to the proof of Theorem 2. That is, we show the existence of a distribution $\mathcal{D} \in \{\mathcal{D}_\ell\}_{\ell \in \{-1,1\}^u}$ such that with a constant probability over $S \sim \mathcal{D}^m$ there exists some voting classifier $f_S \in C(\hat{\mathcal{H}})$ such that $f_S$ has large margins on points in $S$, but has large generalization probability with respect to $\mathcal{D}$. To this end, let $m$ be such that $\frac{\ln N}{\theta^2} < \left( \frac{m}{\lg m} \right)^{9/10}$, and note that $m = \left( \frac{\ln N}{\theta^2} \right)^{1+\Omega(1)}$. Let $u = \frac{40m}{\lg m}$, and let $d = \frac{\ln N}{\theta^2}$.

Similarly to the proof of Theorem 1, the main challenge is to show the existence of a lebeling that satisfies all desired properties. We draw the reader's attention to the fact that unlike the previous proof, the distribution over labelings is not uniform over the entire set $\{-1, 1\}^u$, but rather a designated subset of sparse labelings.

With every labeling $\ell \in \{-1, 1\}^u$ and an $m$-point sample $S$, we associate a classifier $h_{\ell,S}$ as follows. Intuitively, $h_{\ell,S}$ "adversarially changes" at most $d$ labels of points in $\{\xi_2, \ldots, \xi_{u-d}\}$ that were not picked by $S$, and chooses the majority label for points in $\{\xi_{u-d+1}, \ldots, \xi_u\}$. Formally, let $\mathcal{I}_S \subseteq \{\xi_2, \ldots, \xi_{u-d}\} \setminus S$ be an arbitrary sets of size at most $d$, then for every $x \in \{\xi_1, \ldots, \xi_{u-d}\}$, $h_{\ell,S}(x) = -\ell(x)$ if and only if $x \in \mathcal{I}_S$, and for every $x \in \{\xi_{u-d+1}, \ldots, \xi_u\}$, $h_{\ell,S}(x)$ is the majority of labels of $x$ in $S$. That is $h_{\ell,S}(x) = 1$ if and only if $(x, 1)$ appears in $S$ more times than $(x, -1)$. Break ties arbitratily.

**Lemma 9.** *If $\alpha \leq \sqrt{\frac{d}{40\beta m}}$ then there exists $\hat{\ell} \in \{-1, 1\}^u$ such that*

1. *For every $i \in [u - d]$, $\hat{\ell}_i = 1$;*

2. *With probability at least $99/100$ over the choice of sample $S \sim \mathcal{D}_{\hat{\ell}}^m$, there exists a voting classifier $f_S \in C(\hat{\mathcal{H}})$ such that $f_S(\xi_i) h_{\hat{\ell},S}(\xi_i) \geq \theta$ for all $i \in [u]$; and*

3. *with probability at least $1/25$ over $S \sim \mathcal{D}_{\hat{\ell}}^m$ we have*

$$\Pr_{(x,y)\sim\mathcal{D}_{\hat{\ell}}} [y h_{\hat{\ell},S}(x) < 0] \geq \frac{(1-\alpha)\beta}{2} + \frac{(1-\beta)\varepsilon d}{8(u-d-1)} + \frac{\alpha\beta}{24} \,.$$

We first show that the lemma implies Theorem 2.

*Proof of Theorem 2.* Fix some $\tau \in [0, 49/100]$. Assume first that $\tau \leq \frac{d}{50u}$, and let $\varepsilon = \frac{1}{2}$ and $\beta = \alpha = 0$. With probability $1/25$ over $S$ we have

$$\Pr_{(x,y)\sim\mathcal{D}_{\hat{\ell}}} [y h_{\hat{\ell},S}(x) < 0] \geq \frac{(1-\beta)\varepsilon d}{8u} \geq \tau + \Omega \left( \frac{d}{u} \right) = \tau + \Omega \left( \frac{\ln |\hat{\mathcal{H}}| \ln m}{m\theta^2} + \sqrt{\frac{\tau \ln |\hat{\mathcal{H}}| \ln m}{m\theta^2}} \right) \,,$$

where the last transition is due to the fact that $d = \theta^{-2} \ln N = \Theta(\theta^{-2} \ln |\hat{\mathcal{H}}|)$ and $\tau = O(d/u)$. Moreover, with probability $99/100$ over $S$ there exists $f_S \in C(\hat{\mathcal{H}})$ such that $f_S(\xi_i) h_{\hat{\ell},S}(\xi_i) \geq \theta$ for all $i \in [u]$. We get that with probability at least $1/100$ over the sample $S$ there exists $f_S \in C(\hat{\mathcal{H}})$ such that

$$\Pr_S [y_j f_S(x_j) < \theta] = \Pr_S [y_j h_{\hat{\ell},S}(x_j) < 0] = 0 \leq \tau \,,$$

and moreover

$$\Pr_{(x,y)\sim\mathcal{D}_{\hat{\ell}}}[yf_S(x) < 0] = \Pr_{(x,y)\sim\mathcal{D}_{\hat{\ell}}}[yh_{\hat{\ell},S}(x) < 0] \geq \tau + \Omega\left(\frac{\ln|\hat{\mathcal{H}}|\ln m}{m\theta^2} + \sqrt{\frac{\tau\ln|\hat{\mathcal{H}}|\ln m}{m\theta^2}}\right).$$

Otherwise, assume $\tau > \frac{d}{50u}$, and let $\varepsilon = \frac{1}{2}$, $\alpha = \sqrt{\frac{d}{2560\tau m}}$ and $\beta = \frac{64\tau}{32-31\alpha}$. Since $\tau \geq \frac{d}{50u}$, then $\alpha \in [0,1]$. Moreover, for large enough constant $C > 0$, if $m > Cd$, then $32 - 31\alpha \geq 64 \cdot \frac{499}{1000} \geq 64 \cdot \frac{101}{100}\tau$, and therefore $\beta \in [0, 100/101]$.

Next, let $\langle(x_1,y_1),\ldots,(x_m,y_m)\rangle \sim \mathcal{D}_{\hat{\ell}}^m$ be a sample of $m$ points drawn independently according to $\mathcal{D}_{\hat{\ell}}$. For every $j \in [m]$, let $\mathcal{E}_j$ be the event that $(x_j, y_j) \in \{(\xi_i, -\hat{\ell}_i)\}_{i \in [u-d+1,u]}$, then we have $\mathbb{1}_{y_j f_S(x_j) < 0} < \mathbb{1}_{\mathcal{E}_j}$. Moreover, $\mathbb{E}[\mathbb{1}_{\mathcal{E}_j}] = \frac{(1-\alpha)\beta}{2}$, and $\{\mathbb{1}_{\mathcal{E}_j}\}_{j \in [m]}$ are independent. Therefore by Chernoff we get that for large enough $N$,

$$\Pr_{S\sim\mathcal{D}_{\hat{\ell}}^m}\left[\Pr_{(x,y)\sim S}\left[yh_{\hat{\ell},S}(x) < 0\right] \geq \tau\right] \leq \Pr_{S\sim\mathcal{D}_{\hat{\ell}}^m}\left[\frac{1}{m}\sum_{j\in[m]}\mathbb{1}_{\mathcal{E}_j} \geq \frac{(1-31\alpha/32)\beta}{2}\right]$$

$$\leq e^{-\Theta(\alpha^2\beta m)} = e^{-\Theta(d)} \leq 10^{-3},$$

where the inequality before last is due to the fact that $\alpha^2\beta m = \frac{d\beta}{2560\tau} = \Omega(d)$, since $\beta \geq 2\tau$.

Moreover, since $\alpha \leq 1$ then $\beta \leq 64\tau$, and therefore $\alpha = \sqrt{\frac{d}{2560\tau m}} \leq \sqrt{\frac{d}{40\beta m}}$. Thus with probability at least $1/25$ over $S$ we get that

$$\Pr_{(x,y)\sim\mathcal{D}_{\hat{\ell}}}[yh_{\hat{\ell},S}(x) < 0] \geq \frac{(1-\alpha)\beta}{2} + \frac{(1-\beta)\varepsilon d}{u-d-1} + \frac{\alpha\beta}{32} = \frac{(1-31\alpha/32)\beta}{2} + \frac{(1-\beta)\varepsilon d}{u-d-1} + \frac{\alpha\beta}{64}$$

$$= \tau + \Omega\left(\frac{d}{u} + \sqrt{\frac{\tau d}{m}}\right) \geq \tau + \Omega\left(\frac{\ln|\hat{\mathcal{H}}|\ln m}{m\theta^2} + \sqrt{\frac{\tau\ln|\hat{\mathcal{H}}|}{m\theta^2}}\right),$$

Therefore with probability at least $1/50$ over the sample $S$ we get that $\Pr_{(x,y)\sim S}\left[yh_{\hat{\ell},S}(x) < 0\right] \leq \tau$ and moreover

$$\Pr_{(x,y)\sim\mathcal{D}_{\hat{\ell}}}[yh_{\hat{\ell},S}(x) < 0] \geq \tau + \Omega\left(\frac{\ln|\hat{\mathcal{H}}|\ln m}{m\theta^2} + \sqrt{\frac{\tau\ln|\hat{\mathcal{H}}|}{m\theta^2}}\right).$$

Finally, from Lemma 9 and similarly to the first part of the proof, we get that with probability $1/100$ over the choice of $S$ there exists $f_S \in C(\hat{\mathcal{H}})$ such that $h_{\hat{\ell},S}(\xi_i)f_S(\xi_i) \geq \theta$ for all $i \in [u]$. For all these samples $S$ we get that $\Pr_{(x,y)\sim S}[yf_S(x) < \theta] = \Pr_{(x,y)\sim S}\left[yh_{\hat{\ell},S}(x) < 0\right] \leq \tau$ and moreover

$$\Pr_{(x,y)\sim\mathcal{D}_{\hat{\ell}}}[yf_S(x) < 0] = \Pr_{(x,y)\sim\mathcal{D}_{\hat{\ell}}}[yh_{\hat{\ell},S}(x) < 0] \geq \tau + \Omega\left(\frac{\ln|\hat{\mathcal{H}}|\ln m}{m\theta^2} + \sqrt{\frac{\tau\ln|\hat{\mathcal{H}}|}{m\theta^2}}\right).$$

$\square$

For the rest of the section we therefore prove Lemma 9. As with the proof of Lemma 6, we start by lower bounding the expected value of $\Psi_1(\ell, h_{\ell,S}) + \Psi_2(\ell, h_{\ell,S})$ over a choice of a labeling $\ell$ and samples $S \in \mathcal{D}_\ell$. We consider next the subset $\mathcal{L}'$ of $\mathcal{L}(u,d)$ containing all labelings $\ell$ satisfying $\ell_i = 1$ for all $i \in [u]$. Intuitively, by a coupon-collector like argument we show that with very high probability over the sample $S$, there are at least $d$ points in $\{\xi_i\}_{i \in [u-d]}$ not sampled into $S$. The argument lower bounding $\Psi_2$ is identical to the one in the proof of Lemma 9.

**Claim 10.** *If $\alpha \leq \sqrt{\frac{d}{40\beta m}}$ then*

$$\mathbb{E}_{\ell \in \mathcal{L}'} \left[ \mathbb{E}_S \left[ \Psi_1(\ell, h_{\ell,S}) + \Psi_2(\ell, h_{\ell,S}) \right] \right] \geq \frac{(1-\varepsilon)\beta d}{2(u-d-1)} + \frac{\alpha\beta}{6} .$$

*Proof.* Let $\mathcal{S}$ be the set of all $m$-point samples $S$ for which $|\{\xi_2, \ldots, \xi_{u-d}\} \setminus S| \geq d$. For every $S \in \mathcal{S}$ we have $|\mathcal{I}_S| = d$, and therefore

$$\sum_{i \in [2, u-d]} \mathbb{1}_{\ell_i f_S(\xi_i) < 0} = \sum_{i \in [2, u-d]} \mathbb{1}_{f_S(\xi_i) < 0} = |\mathcal{I}_S| = d .$$

Therefore $\mathbb{E}_\ell[\mathbb{E}_S[\Psi_1(\ell, f_S) | S \in \mathcal{S}]] = \frac{(1-\varepsilon)\beta d}{u-d-1}$. We will show next that $\Pr_S[\mathcal{S}] \geq 1/2$, and conclude that $\mathbb{E}_\ell[\mathbb{E}_S[\Psi_1(\ell, f_S)]] \geq \frac{(1-\varepsilon)\beta d}{2(u-d-1)}$. To see this, consider a random sampling $S \sim \mathcal{D}_\ell^m$. We will show by a coupon-collector argument that with high probability, no more than $(u - d - 1) - d$ elements of $\{\xi_2, \ldots, \xi_{u-d}\}$ are sampled to $S$, and therefore $S \in \mathcal{S}$. Consider the set of elements of $\{\xi_2, \ldots, \xi_{u-d}\}$ sampled by $S$. For every $k \in [u - 2d - 1]$, let $X_k$ be the number of samples between the time $(k-1)$th distinct element was sampled from $\{\xi_2, \ldots, \xi_{u-d}\}$ and the time the $k$th distinct element was sampled from $\{\xi_2, \ldots, \xi_{u-d}\}$. Then $X_k \sim Geom(p_k)$, where $p_k = (1-\beta)\varepsilon \cdot \frac{u-d-k}{u-d-1}$. Denote $X := \sum_{k \in [u-2d-1]} X_k$, then

$$\mathbb{E}[X] = \sum_{k \in [u-2d-1]} \frac{1}{p_k} = \sum_{k \in [u-2d-1]} \frac{u-d}{(1-\beta)\varepsilon(u-d-k)} = \frac{u-d-1}{(1-\beta)\varepsilon} \sum_{k=d+1}^{u-d-1} \frac{1}{k}$$

$$\geq (u-d-1)[\ln(u-d-1) - \ln(d+1) - 1] \geq \frac{1}{2} u \ln \frac{u}{d} \geq \frac{1}{20} u \ln u \geq \frac{4}{3} m$$

Therefore by letting $\lambda = \frac{3}{4}$, and $p_* = \min_{k \in [u-2d-1]} p_k = (1-\beta)\varepsilon \cdot \frac{u-d-(u-2d-1)}{u-d-1} \geq \frac{d}{u}$ then known tail bounds on the sum of geometrically-distributed random variable (e.g. [Jan18, Theorem 3.1]) we get that for large enough values of $m$,

$$\Pr_{S \sim \mathcal{D}^m}[S \notin \mathcal{S}] = \Pr[X \leq m] \leq \Pr[X \leq \lambda \mathbb{E}[X]] \leq e^{-p_* \mathbb{E}[X](\lambda-1-\ln\lambda)} \leq e^{-\Omega(\ln u)} \leq 1/2 . \quad (10)$$

The lower bound on the expectation of $\Psi_2$ is proved identically to the proof in Claim 7. $\qquad\square$

Similarly to Claim 8, we conclude the following.

**Claim 11.** *For $\alpha \leq \sqrt{\frac{d}{40\beta m}}$, then with probability at least $1/11$ over the choice of $\ell \in \mathcal{L}'$ we have*

$$\Pr_{S \sim \mathcal{D}_\ell^m} \left[ \Psi_1(\ell, h_{\ell,S}) + \Psi_2(\ell, h_{\ell,S}) \geq \frac{(1-\beta)\varepsilon d}{4(u-d-1)} + \frac{\alpha\beta}{12} \right] \geq \frac{1}{25} .$$

We next want to show that there exists a labeling $\ell \in \mathcal{L}'$ such that with high probability over $S \sim \mathcal{D}_\ell^m$, there exists a voting classifier $f_S \in C(\hat{\mathcal{H}})$ attaining high margins with $h_{\ell,S}$. since the distribution induced on $\{\xi_i\}_{i \in [u-d+1,u]}$ by $\mathcal{D}_\ell$ is uniform, we conclude the following for a large enough value of $N$.

**Claim 12.** *With probability at least $99/100$ over the choice of a labeling $\ell \in \mathcal{L}'$,*

$$\Pr_{S \sim \mathcal{D}_\ell} \left[ \exists f_S \in C(\hat{\mathcal{H}}) : \forall i \in [i]. \ h_{\ell,S}(\xi_i) f_S(\xi_i) \geq \theta \right] \geq \frac{99}{100} .$$

*Proof.* For two labelings $\ell \in \mathcal{L}(u,d)$ and $\ell' \in \mathcal{L}'$ we say that $\ell$ and $\ell'$ are similar, and denote $\ell \equiv \ell'$ if for all $i \in [u - d + 1, u]$, $\ell_i = \ell_i'$. From Claim 5 we know that

$$1 - 1/N \leq \Pr_{\ell \in_R \mathcal{L}(u,d)} [\exists f \in C(\hat{\mathcal{H}}) : \forall i \in [u]. \ \ell_i f(\xi_i) \geq \theta] =$$

$$= \sum_{\ell' \in \mathcal{L}} \Pr_{\ell \in_R \mathcal{L}(u,d)} [\exists f \in C(\hat{\mathcal{H}}) : \forall i \in [u]. \ \ell_i f(\xi_i) \geq \theta | \ell \equiv \ell'] \cdot \Pr_{\ell \in_R \mathcal{L}(u,d)} [\ell \equiv \ell']$$

$$= \sum_{\ell' \in \mathcal{L}} \Pr_{S \sim \mathcal{D}_{\ell'}^m} [\exists f_S \in C(\hat{\mathcal{H}}) : \forall i \in [u]. \ h_{\ell',S}(\xi_i) f(\xi_i) \geq \theta | \ell \equiv \ell'] \cdot \Pr_{\ell \in_R \mathcal{L}(u,d)} [\ell \equiv \ell']$$

For a large enough value of $N$ we conclude that with probability at least $99/100$ over a choice of $\ell' \in \mathcal{L}'$, for at least a $99/100$ fraction of samples $S \sim \mathcal{D}_{\ell'}^m$ there exists a voting classifier $f_S \in C(\hat{\mathcal{H}})$ attaining high margins with $h_{\ell',S}$. $\qquad\square$

Combining Claims 12 and 11 we conclude that if $\alpha \leq \sqrt{\frac{d}{40\beta m}}$ then there exists $\hat{\ell} \in \mathcal{L}'$ satisfying the guarantees in Lemma 9. The proof of the lemma, and therefore of Theorem 2 is now complete.

# 4   Existence of a Small Hypotheses Set

Fix some $\theta \in (0, 1/40)$, $\delta \in (0, 1)$ and an integer $d \leq u$. Let $\gamma = 4\theta \in (0, 1/10)$ and let $N = 2\gamma^{-2} \ln d \cdot \ln \frac{\gamma^{-2} \ln d}{\delta} \cdot e^{O(\theta^2 d)}$. We define the distribution $\mu$ via the following procedure, that samples a hypothesis set $\mathcal{H} \sim \mu$. Let $\hat{h} : \mathcal{X} \to \{-1, 1\}$ be defined by $\hat{h}(x) = 1$ for all $x \in \mathcal{X}$. Sample independently and uniformly at random $N$ hypotheses $h_1, \ldots, h_N \in_R \mathcal{X} \to \{-1, 1\}$, and define $\mathcal{H} := \{\hat{h}\} \cup \{h_j\}_{j \in [N]}$.

Clearly every $\mathcal{H} \in \text{supp}(\mu)$ satisfies $|\mathcal{H}| = N + 1$. We therefore turn to prove the second property. To this end, let $k = \gamma^{-2} \ln d$. In order to show existence of a voting classifier, we conceptually change the procedure defining $\mu$, and think of the random hypotheses as being sampled in $k$ equally sized "batches", each of size $N/k$, and adding $\hat{h}$ to each of them. Denote the batches by $\mathcal{H}_1, \mathcal{H}_2, \ldots, \mathcal{H}_k$. We consider next the following procedure to construct a voting classifier $f \in C(\mathcal{H})$ given $\mathcal{H} \sim \mu$. We will use the main ideas from the AdaBoost algorithm. Recall that AdaBoost creates a voting classifier using a sample $S = ((x_1, y_1), \ldots, (x_u, y_u))$ in iterations. Staring with $f_0 = 0$, in iteration $j$, it computes a new voting classifier $f_j = f_{j-1} + \alpha_j h_j$ for some hypothesis $h_j \in \mathcal{H}$ and weight $\alpha_j$. The heart of the algorithm lies in choosing $h_j$. In each iteration, AdaBoost computes a distribution $D_j$ over $S$ and chooses a hypothesis $h_j$ minimizing

$$\varepsilon_j = \Pr_{i \sim D_j}[h_j(x_i) \neq y_i].$$

The weight it then assigns is $\alpha_j = (1/2) \ln((1 - \varepsilon_j)/\varepsilon_j)$ and the next distribution $D_{j+1}$ is

$$D_{j+1}(i) = \frac{D_j(i) \exp(-\alpha_j y_i h_j(x_i))}{Z_j}$$

where $Z_j$ is a normalization factor, namely

$$Z_j = \sum_{i=1}^{d} D_j(i) \exp(-\alpha_j y_i h_j(x_i)).$$

The first distribution $D_1$ is the uniform distribution.

We alter the above slightly assigning uniform weights on the hypotheses, and setting $\alpha_j = \frac{1}{2} \ln \frac{1+2\gamma}{1-2\gamma}$ for all iterations $j$. The algorithm is formally described as Algorithm 1.

We will prove that the algorithm fails with probability at most $\delta$ (over the choice of $\mathcal{H}$), and that if the algorithm does not fail, then it returns a voting classifier with minimum margin at least $\theta$. First note that if $f$ is the classifier returned by the algorithm, then clearly $f = \frac{1}{k} \sum_{j \in [k]} h_j \in C(\mathcal{H})$ is a voting classifier.

**Claim 13.** *Algorithm 1 fails with probability at most $\delta$.*

*Proof.* Since $\mathcal{H}_1, \ldots, \mathcal{H}_k$ are independent, it is enough to show that for every $j \in [k]$, for every $w \in \Delta_u$ with probability at least $1 - \delta/k$ there exsits $h_j \in \mathcal{H}_j$ such that

$$\sum_{i \in [u]} w_i \mathbb{1}_{y_i \neq h_j(x_i)} \leq \frac{1}{2} - \gamma, \tag{11}$$

where $\Delta_u$ is the $u$-dimensional simplex. First note that if $\sum_{i \in [u]:y_i=-1} w_i \leq \frac{1}{2} - \gamma$, then $\hat{h} \in \mathcal{H}_j$ satisfies (11). We can therefore assume $\sum_{i \in [u]:y_i=-1} w_i > \frac{1}{2} - \gamma$. Next, note that for every

---
**Input:** $(\mathcal{H}_1, \ldots, \mathcal{H}_k) \sim \mu$

**Output:** $f \in C\left(\bigcup_{j \in [k]} \mathcal{H}_j\right)$

1: let $\alpha = \frac{1}{2} \ln \frac{1+2\gamma}{1-2\gamma}$
2: let $f(x) = 0$ for all $x \in \mathcal{X}$
3: let $D_1(i) = \frac{1}{u}$ for all $i \in [u]$.
4: **for** $j = 1$ to $k$ **do**
5:     Find a hypothesis $h_j \in \mathcal{H}_j$ satisfying $\sum_{i \in [u]} D_j(i) \mathbb{1}_{y_i \neq h_j(x_i)} \leq \frac{1}{2} - \gamma$.
       If there is no such hypothesis, **return** *fail*.
6:     $f_j \leftarrow f_{j-1} + h_j$.
7:     $Z_j \leftarrow \sum_{i \in [u]} D_j(i) \exp(-\alpha y_i h_j(x_i))$.
8:     for every $i \in [u]$ let $D_{j+1}(i) = \frac{1}{Z_j} D_j(i) \exp(-\alpha y_i h_j(x_i))$.
9: **return** $\frac{1}{k} f_k$.
---

**Algorithm 1:** Construct a Voting Classifier

508     $h : \mathcal{X} \to \{-1, 1\}$ we have

$$\sum_{i \in [u]} w_i \mathbb{1}_{y_i \neq h(x_i)} = \sum_{i \in [u]} \frac{1}{2}(w_i - w_i y_i h(x_i)) = \frac{1}{2}\left(\sum_{i \in [u]} w_i - \sum_{i \in [u]} w_i y_i h(x_i)\right) = \frac{1}{2} - \frac{1}{2}\sum_{i \in [u]} w_i y_i h(x_i)$$

509     Therefore $\sum_{i \in [u]} w_i \mathbb{1}_{y_i \neq h(x_i)} \geq \frac{1}{2} - \gamma$ if and only if $\sum_{i \in [u]} w_i y_i h(x_i) \geq 2\gamma$. We want to show
510     that with probability at most $\frac{\delta}{k}$ every $h \in \mathcal{H}_j$ satisfies $\sum_{i \in [u]} w_i y_i h_j(x_i) \geq 2\gamma$. We claim that it is
511     enough to show that

$$\Pr_{h \in_R \mathcal{X} \to \{-1,1\}}\left[\sum_{i \in [u]} w_i y_i h(x_i) \geq 2\gamma\right] \geq \frac{k \ln \frac{k}{\delta}}{N} = \frac{1}{2} e^{-\Theta(\gamma^2 d)} \tag{12}$$

512     To see why this is enough assume that (12) is true, then since sampling $\mathcal{H}_j$ means indepently and
513     uniformly sampling $N/k$ hypotheses $h \in_R \mathcal{X} \to \{-1, 1\}$, the probability that there exists $h \in \mathcal{H}_j$
514     such that (11) holds is at least

$$1 - (1 - \frac{k \ln \frac{k}{\delta}}{N})^{N/k} \geq 1 - \exp\left(-\frac{k \ln \frac{k}{\delta}}{N} \cdot \frac{N}{k}\right) = 1 - \frac{\delta}{k}.$$

515     We thus turn to prove that (12) holds. To this end, let $M := \{i \in [u] : \beta_i < 0\}$. Recall that
516     $|M| \leq d$ and that we assumed $\sum_{i \in M} w_i = \sum_{i \in M} |y_i w_i| \geq \frac{1}{2} - \gamma$. From a known tail bound
517     by Montgomery-Smith [MS90] on the sum of Rademacher random variables we have that since
518     $\gamma \in (0, 1/10)$,

$$\Pr\left[\sum_{i \in [u]} w_i y_i h(x_i) \geq 2\gamma\right] \geq \Pr\left[\sum_{i \in M} w_i y_i h(x_i) \geq 2\gamma \text{ and } \sum_{i \in [u] \setminus M} w_i y_i h(x_i) \geq 0\right] \geq \frac{1}{2} e^{-\Theta(\gamma^2 d)}$$

519                                                                                                                                 $\square$

520     **Claim 14.** *If Algorithm 1 does not fail, then for every $i \in [y]$, $y_i f(x_u) \geq \theta$.*

521     *Proof.* We first show by induction that for all $j \in [k]$ we have that for all $i \in [u]$

$$\exp(-\alpha y_i f_j(x_i)) = u \cdot D_{j+1}(i) \prod_{\ell \in [j]} Z_\ell .$$

522     To see this observe that for all $i \in [u]$, $D_2(i) = \frac{D_1(i)}{Z_1} \exp(-\alpha y_i h_1(x_i))$. Since $h_1 = f_1$ and by
523     rearranging we get that $\exp(-\alpha y_i f_1(x_i)) = \frac{D_2(i)Z_1}{D_1(i)} = u \cdot D_2(i)Z_1$. For the induction step we have

that
$$\exp(-\alpha y_i f_j(x_i)) = \exp(-\alpha y_i(f_{j-1}(x_i) + h_j(x_i))) = \exp(-\alpha y_i f_{j-1}(x_i)) \cdot \exp(-\alpha y_i h_j(x_i))$$
$$= u \cdot D_j(i) \prod_{\ell \in [j-1]} Z_\ell \cdot \frac{Z_j D_{j+1}(i)}{D_j(i)}$$
$$= u \cdot D_{j+1}(i) \prod_{\ell \in [j]} Z_\ell$$

Since $\sum_{i \in [u]} D_{k+1}(i) = 1$, we get that
$$\sum_{i \in [u]} \exp(-\alpha y_i f_k(x_i)) = u \prod_{\ell \in [k]} Z_\ell . \tag{13}$$

We turn therefore to bound $Z_\ell$ for $\ell \in [k]$. Denote $\varepsilon_\ell = \sum_{i \in [u]} D_\ell(i) \cdot \mathbb{1}_{h_\ell(x_i) \neq y_i}$. Then
$$Z_\ell = \sum_{i \in [u]} D_\ell(i) \exp(-\alpha y_i h_\ell(x_i)) = \sum_{i \in [u]} D_\ell(i) \exp\left(-\frac{1}{2} \ln\left(\frac{1+2\gamma}{1-2\gamma}\right) y_i h_\ell(x_i)\right)$$
$$= \sum_{i \in [u]} D_\ell(i) \left(\frac{1+2\gamma}{1-2\gamma}\right)^{-\frac{1}{2} y_i h_\ell(x_i)} = \varepsilon_\ell \left(\frac{1+2\gamma}{1-2\gamma}\right)^{\frac{1}{2}} + (1-\varepsilon_\ell)\left(\frac{1+2\gamma}{1-2\gamma}\right)^{-\frac{1}{2}}$$
$$= \left(\frac{\varepsilon_\ell}{1-2\gamma} + \frac{1-\varepsilon_\ell}{1+2\gamma}\right)\sqrt{(1+2\gamma)(1-2\gamma)}$$

By the condition in line 5 we know that $\varepsilon_\ell \leq \frac{1}{2} - \gamma$. Since $\left(\frac{\varepsilon_\ell}{1-2\gamma} + \frac{1-\varepsilon_\ell}{1+2\gamma}\right)$ is increasing as a function of $\varepsilon_\ell$ we therefore get that
$$Z_\ell \leq \left(\frac{\frac{1}{2}-\gamma}{1-2\gamma} + \frac{\frac{1}{2}+\gamma}{1+2\gamma}\right)\sqrt{(1+2\gamma)(1-2\gamma)} = \sqrt{(1+2\gamma)(1-2\gamma)} \leq 1 - 2\gamma^2 ,$$

where the last inequality follows from the fact that $1 - 4\gamma^2 \leq (1-2\gamma^2)^2$. Substituting in (13) we get that for every $i \in [u]$,
$$\exp(-\alpha y_i f_k(x_i) \leq \sum_{i \in [u]} \exp(-\alpha y_i f_k(x_i)) = u \prod_{\ell \in [k]} Z_\ell \leq u \cdot \left(1 - 2\gamma^2\right)^k \leq \exp(\ln d - 2k\gamma^2) ,$$

and therefore
$$y_i f(x_i) = \frac{1}{k} y_i f_k(x_i) \geq \frac{1}{k\alpha}(2k\gamma^2 - \ln d) . \tag{14}$$

Since $\ln(1+x) \leq x$ for all $x \geq 0$ we get that
$$\alpha = \frac{1}{2}\ln\left(\frac{1+2\gamma}{1-2\gamma}\right) = \frac{1}{2}\ln\left(1 + \frac{4\gamma}{1-2\gamma}\right) \leq \frac{2\gamma}{1-2\gamma} \leq 4\gamma ,$$

where the last inequality follows from the fact that $\gamma \in (0, 1/4)$. Substituting in (14) we get that
$$y_i f(x_i) \geq \frac{1}{4k\gamma}(2k\gamma^2 - \ln d) = \frac{\gamma}{2} - \frac{\ln d}{4k\gamma} .$$

Recall that $k = \gamma^{-2}\ln d$, and therefore $y_i f(x_i) \geq \gamma/4 = \theta$. $\qquad\square$

## 5  Conclusions

In this work, we showed almost tight margin-based generalization lower bounds for voting classifiers. These new bounds essentially complete the theory of generalization for voting classifers based on margins alone. Closing the remaining gap between the upper and lower bounds is an intriguing open problem and we hope our techniques might inspire further improvements. Our results come in the form of two theorems, one showing generalization lower bounds for *any* algorithm producing a voting classifier, and a slightly stronger lower bound showing the *existence* of a voting classifier with poor generalization. This raises the important question of whether specific boosting algorithms can produce voting classifiers that avoid the $\lg m$ factor in the second lower bound via a careful analysis tailored to the algorithm. As a final important direction for future work, we suggest investigating whether natural parameters other than margins may be used to better explain the practical generalization error of voting classifiers. At least, we now have an almost tight understanding, if no further parameters are taken into consideration.