[Reviews · NeurIPS 2019]

Reviewer 1



The paper is well written and even though notation might be heavy when reading the proofs, the authors try to give intuition behind their approach. I have minor questions about some of the presentation in the paper. On line 167 do the authors assume that the size of the population is |\mathcal{X}| = 10m? On line 176 shouldn’t the inequality be an equality, otherwise how is the above distribution over \mathcal{X} proper? While first reading the paper, I had the following confusion about the intuitive explanation between lines 179 and 181: we need the mass on the single point to be very large compared to m, however, in that case it is highly likely that the returned classifier will have small generalization error because it will classify the point with large mass correctly and then the probability to sample every other point upper bounds the generalization error. Maybe the authors can hint at what the size of \mathcal{X} is compared to m? From my understanding an important part of the lower bound proof is carefully balancing between the distribution which puts large mass on a single point and the distribution which slightly biases one of the labels compared to the other. It still a bit mysterious why exactly these two types of distributions are needed. It seems that the terms appearing in Eq. 3 are the ones governed by the two distributions. Elaborating more on why these terms show up and their relation to the two distributions might make the presentation better. Overall the contributions in terms of lower bounds are significant and almost match the state of the art known upper bounds. The proofs are non-trivial and to my knowledge are novel.

Reviewer 2



This paper provides a lower bound that matches existing margin-based upper bounds for voting classifiers. These existing upper bounds bound the true error of a voting classifier, as generated by boosting algorithms, by the sum of its empirical probability of obtaining a large margin, and a generalization term that depends on this margin. The provided lower bound nearly matches this upper bound, up to log factors and small powers. The paper presents two theorems. Theorem 2 is a lower bound that can be directly compared to the upper bound. Theorem 1, in contrast, replaces the empirical margin of the voting classifier returned by an algorithm (f_{A,S}) with the empirical margin of another voting classifier (f). This change of classifiers results in a lower bound that does not have a direct bearing to the upper bound, which is phrased in terms of a single voting classifier. The discussion of Theorem 1 (see line 113) doesn't seem to acknowledge this difference or explain why this result can still be considered a matching lower bound. Theorem 2 does provide a lower bound which is directly comparable to known upper bounds, and is tight up to log factors and small powers. This theorem only holds for sample sizes that are slightly larger than what would be required to avoid triviality, m = (ln N/\theta^2)^(1+\Omega(1)). Strangely, this limitation is completely ignored in the text, although other easier limitations are discussed. The authors should address this limitation explicitly in their description of the result. The notation \Omega(1) in the power of m should be replaced by the actual number: the value here is of importance, since large values would make the result much weaker. It seems, from the supplementary material, that the actual value might be reasonable, however it should be explicitly spelled out. Almost none of the proofs are provided in the body of the paper. Instead, long explanations describing the proof parts that are based on standard lower-bounding techniques are provided. This is unfortunate, since the supplementary material seems to include some interesting proofs that could have been easily incorporated in the body of the paper instead of long explanations of the standard parts of the proof. The result is a paper which provides almost no interesting technical contribution, although such contribution does seem to exist --- in the supplementary material. The most interesting result and proof seem to be Lemma 3, which shows the existence of a relatively small set of hypotheses that admit voting classifiers with a given minimum margin if only a small number of the labels are negatively labeled. The proof of this lemma uses an interesting construction technique based on AdaBoost. However, like all other proofs, it is not presented in the body of the paper. Considering the significance and novelty of the result, I find the provided lower bound mildly interesting, though not surprising. Had the proof of Lemma 3 been provided in the body of the paper, along with some of the more interesting parts of the rest of the proofs, this might have made a more interesting contribution. The current organization leaves most of the actual contribution out of the paper. In fact, very little that can be technically verified is provided in the body of the paper and almost all the derivations are in the supplementary material. - line 110 "it is always possible" -- this should only be possible with the D whose existence is proved in theorem 1. - Sometimes lg is used, and sometimes ln. Please be consistent. - line 407 in the supplementary: "lebeling"-->"labeling" - Reference SFB+ includes an "et al" which is out of place. - The mention that proofs are "deferred to the full version of this work" repeats in many places, where in fact the proofs are provided in the supplementary material.

Reviewer 3



The authors provide (almost tight) lower bounds for the generalization errors for boosting based classifiers, in terms of the empirical margin, thus showing that the algorithm due to GZ13 is almost optimal when the only problem parameter is the margin. This work, hence, encourages a search for other data-dependent empirical quantities for the problem exhibiting stronger generalization. It is good to understand the problem well, but I fear that I do not see a lot of impact of this result on the way people currently think about boosting. All their lower bounds, construct a distribution D (which may depends on m, H and A). This is the standard flavor of lower bounds for the generalization error. However, the lower bounds developed are not fully satisfying. Theorem 1 does not suggests that the returned classifier which has a large test error, has small \theta-margin error on the training data. Theorem 2 does not suggest that the presented f which satisfies both the properties is the one returned by any “interesting” algorithm or is even computable from data. An ideal lower bound should look like - “No algorithm can return a classifier f which has small error (margin) on the training set, but a large test error for all data-distributions D.” Originality: The authors combine old ideas to prove lower bounds under different settings in a very clever way to get the required lower bound. The idea of lower bounding the population risk by the sum term (claim 4 - supplementary doc), which is further easier to lower bound with high probability, is also quite interesting. However, the techniques used to prove the lower bounds in each of the pieces are not new. Writing: The paper is overall well written. Using the same variable d to represent both the sparsity parameter and the data-set split (in lines 274-285 in the supplementary) is confusing, and I would recommend the authors to fix it. Line 289 also contains a small typo in the definition of \psi_1. ** After Author Feedback: Thank you for clarifying my confusions regarding Theorem 2. I find it quite interesting now. I would like to keep the same scores as Theorem 1 is still quite unsatisfying and the author responses did not really help.

[Author Response · NeurIPS 2019]

**Author Response ("Margin-Based Generalization Lower Bounds for Boosted Classifiers")**

We thank the reviewers for the time and expertise invested in these reviews.

**Response to the First Reviewer (Reviewer #3).** Regarding the intuition behind the "first part" of the distribution
(lines 164-179): Thanks for pointing out the confusion, we will try to make the presentation more precise. The intuition
we wanted to get across was the following: Assume we could assign a probability of $\frac{1}{10m}$ to every point in $\mathcal{X}$ (which
as you say require $|\mathcal{X}| = 10m$). Then most data points would not be sampled. This is great for proving a high
generalization error: On a sample of $m$ points, one would only see a small constant fraction of $\mathcal{X}$ and the error would be
about $1/2$. Now the issue with the above is that the sample $S$ will consist of $m$ distinct points with a randomly chosen
label. This makes it impossible to construct a small hypothesis set that can guarantee a good margin on the sample
(point 1. of Theorem 1). Thus instead we create only $d$ points with a probability of being sampled of $1/10m$. Of course
the distribution is not proper now (as you also remark), hence we need to add one point with large mass. Since a sample
will miss a constant fraction of the $d$ points, the generalization error will be proportional to $d/m$. The last part of the
proof is choosing the largest $d$ for which we can still guarantee good margins on the sample. It turns out we can choose
$d$ (and hence $|\mathcal{X}|$) as $\theta^{-2} \ln |\mathcal{H}|$. We will make sure to comment on $|\mathcal{X}|$ in the final version.

We will also try to add more intuition about the role of the two distributions in the final version of the paper. Intuitively,
the second distribution with the slightly biased labels is preferable in terms of proving large generalization error because
it ensures that an algorithm often will be wrong also on points it has already seen in the sample. But we cannot use
that distribution all the time, as it also means that many margins will become negative (when the same point has
been sampled with two different labels, the margin will be negative on one sample). So the proof "uses" the second
distribution as much as the threshold $\tau$ allows, and uses the first distribution the remaining time (which yields better
margins but smaller error).

**Response to the Second Reviewer (Reviewer #4).** Regarding the statement of Theorem 1, the change of classifiers
is essential to give the theorem meaning. And in fact, it makes the theorem even *stronger*. In particular, since Theorem 1
applies *for all algorithms*, it *simultaneously* applies to *every* algorithm that maximizes margins. One could e.g. choose
the algorithm that given a sample spends arbitrarily much time in order to find the voting classifier with the best margins
possible on the sample, and still that algorithm would have a large error. Theorem 1 also applies to all other algorithms,
even those not trying to maximize margins but maybe something completely different. Since we want Theorem 1 to
state something about all possible algorithms, we cannot hope to show that the classifier constructed by the algorithm
performs well on the training set, that is, it has a good empirical margin. Hence we need the change of classifier. On
the other hand, if we discard the first condition altogether, then the claim becomes almost trivial (at least for smaller
values of $\tau$) as a uniform distribution over $\mathcal{X} \times \{-1, 1\}$ can then fail any classifier. We will make sure to emphasize
this further in a final version.

Regarding the constant in the exponent in the definition of $m$ for Theorem 2, the thing is that any constant exponent
bounded away from 1 will do here. To be concrete the proof requires $\frac{m}{\ln m} \geq \left(\frac{\ln N}{\theta^2}\right)^{1+1/10}$, which in turn is used in the
proof of Claim 10 right after line 461 to get that $\ln(u/d) \geq (1/10) \ln u$. However the choice of $1/10$ is arbitrary. This
will be explicitly stated in the final version of the paper.

The main consideration for not including the more technical parts of the proofs was to allow us to provide the reader
with an intuitive high-level description. As can be seen in the full version of the paper, which was submitted as
supplementary material, all the proofs are provided in detail. At the reviewers' discretion, we can include more of the
details in the final version of the paper. In any case, the full version of the paper will be published on arXiv.

Naturally, the typos pointed out by the reviewer will be rectified.

**Response to the Third Reviewer (Reviewer #5).** We believe there is a slight confusion as to the strength of
Theorem 1. In particular, Theorem 1 holds for *every algorithm*. This means that one could e.g. take the algorithm $\mathcal{A}$
that outputs the voting classifier obtaining the best possible margins on the sample, and still that algorithm has large
generalization error. And by the first point in Theorem 1, that algorithm $\mathcal{A}$ can (and thus will) actually have good
margins. Thus Theorem 1 is *even stronger* than if it had said that the any algorithm $\mathcal{A}$ which produces good margins
also has large generalization error. Please also see the response to the Second Reviewer where this is also discussed.

Regarding the statement of Theorem 2, the existence of a classifier that satisfies both properties of the theorem shows
that one cannot rely on high margins on the training set in order to attain performance better than the upper bounds
provide. Thus showing that the known upper bounds are almost tight, if one relies only on the margin distribution.

Once again, we thank all the reviewers for their time and effort invested in this paper, and for valuable remarks.

[Meta-Review · NeurIPS 2019]

The paper presents margin-based lower bounds for boosted classifiers that match the known upper bounds — an exciting result given the popularity and success of boosting methods in machine learning. A typical bound for boosting bounds the generalization error in terms of the empirical margin loss (which increases with the margin) and a complexity term (that decreases with the margin). Therefore, a predictor that achieves small empirical margin loss with a large margin would generalize well. The lower bounds in the paper match the known upper bounds. There was a bit of confusion about the statement of Theorem 1, and authors did well to clarify those doubts in their response. They are strongly encouraged to incorporate their comments into the revised version lest there is any doubt for another reader. In particular, they should emphasize that the existence of a predictor with empirical margin risk bounded by \tau implies that a learning algorithm that minimizes empirical risk will find a predictor whose empirical margin risk is no larger than \tau; if possible, it would be nice to have a formal statement to that effect. In fact, in thinking more about it, I wonder if the risk of a predictor that minimizes the empirical margin loss should not be much smaller than \tau either. Another concern echoed by a reviewer was that many interesting contributions were relegated to the supplementary material and that the paper was merely an advertisement of the results. While it is true that the conference papers are limited by space, I would encourage authors to include more of the proof sketch and novel ideas/insights from the appendix in the main paper.